

# Anthropogenic and catchment characteristic signatures in the water quality of Swiss rivers: a quantitative assessment

Martina Botter[1], Paolo Burlando[1], Simone Fatichi[1]

[1]Institute of Environmental Engineering, ETH Zurich, Switzerland

*Corresponding author*:  Martina Botter,
Institute of Environmental Engineering, ETH Zurich, Switzerland
Stefano Franscini-Platz 5, HIF CO 46.7, 8093 Zurich, Switzerland
Tel.: +41-44-6333992
botter@ifu.baug.ethz.ch

Submitted version to Hydrology and Earth System Sciences

[19 March 2019]

**Abstract**

The hydrological and biogeochemical response of rivers carries information about solute sources, pathways, and transformations in the catchment. We investigate long-term water quality data of eleven Swiss catchments with the objective to discern the influence of major catchment characteristics and anthropic activities on delivery of solutes in stream water. Magnitude, trends, and seasonality of water quality samplings of different solutes are evaluated and compared across catchments. Subsequently, the empirical dependence between concentration and discharge is used to classify the solute behaviors.

While the anthropogenic impacts are clearly detectable in the concentration of certain solutes (i.e., $Na^+$, $Cl^-$, $NO_3$, DRP), the influence of single catchment characteristics as geology (e.g., on $Ca^{2+}$ and $H_4SiO_4$), topography (e.g., on DOC, TOC and TP), and size (e.g., on DOC and TOC) is only sometimes visible, also because of the limited sample size and the spatial heterogeneity within catchments. Solute variability in time is generally smaller than discharge variability and the most significant trends in time are due to temporal variations of anthropogenic rather than natural forcing. The majority of solutes shows dilution with increasing discharge, especially geogenic species, while sediment-bonded solutes (e.g. Total Phosphorous and Organic Carbon species) show higher concentrations with increasing discharge. Both natural and anthropogenic factors affect the biogeochemical response of streams and, while the majority of solutes show identifiable behaviors in individual catchments, only a minority of behaviors can be generalized across the 11 catchments that exhibit different natural, climatic, and anthropogenic features.

**Keywords**: water quality, catchment biogeochemistry, stream chemistry, concentration-discharge relations.

## 1. Introduction

Hydrological and biogeochemical responses of catchments are essential for understanding the dynamics and fate of solutes within the catchment, as material transported with water carries information about water sources, residence time, and biogeochemical transformations [*Abbott et al.*, 2016]. A quantitative description of water quality trends can also shed light on the consequences of anthropogenic changes in the catchment as well as on the possibilities for preventive or remedial actions [*Turner and Rabalais*, 1991]. Concerning changes in watershed land use or management practices, for example, the United States Geological Survey (USGS) established the Hydrologic Benchmark Network (HBN) [*Leopold*, 1962], a long-term monitoring system of dissolved concentrations in 59 differently impacted sites across the United States with the goal of quantifying the human influence on the ecosystems [*Beisecker and Leifeste*, 1975]. Water quality monitoring and assessment are also crucial for stream and catchment restoration, which has been widely practiced in the USA and Europe for several decades and still represent an important challenge of river basin management. However, the system responses to restoration often contradicts a priori expectations, and the lack of adequate monitoring and assessment of basin functioning before the application of restoration measures is considered to be one of the main reasons for this discrepancy [*Hamilton*, 2011].

The relationship between observed in-stream solute concentrations and discharge has been explored in various catchments and with different methods in the last decades [*Langbein and Dawdy*, 1964; *Johnson et al.*, 1969; *Hall*, 1970; *Hall*, 1971; *White and Blum*, 1995; *Evans and Davies*, 1998; *Calmels et al.*, 2011]. One emerging postulate is that concentration ($C$)-discharge ($Q$) relations represent the quantitative expression of the interaction between catchment geomorphology, land use, hydrological processes and the solute releases, thus reflecting in lumped form the complex mixing process taking place along flow paths of variable lengths and residence time [*Chorover et al.*, 2017]. Therefore, *C-Q* relations have been studied with reference to hydrological variables, e.g., hydrologic connectivity and residence time [*Herndon et al.*, 2015; *Baronas et al.*, 2017; *Duncan et al.*, 2017a; *Gwenzi et al.*, 2017; *Torres et al.*, 2017], biological processes [*Duncan et al.*, 2017a], catchment characteristics, e.g., catchment topography, land use, catchment size, and lithological properties [*Musolff et al.*, 2015; *Baronas et al.*, 2017; *Diamond and Cohen*, 2017; *Hunsaker and Johnson*, 2017; *Moatar et al.*, 2017; *Wymore et al.*, 2017], as well as anthropic activities [*Basu et al.*, 2010; *Thompson et al.*, 2011; *Musolff et al.*, 2015; *Baronas et al.*, 2017].

In a log($C$)-log($Q$) space, *C-Q* relations have been observed to be usually linear [*Godsey et al.*, 2009], so that the empirical relations can be well approximated by a power-law, $C = a \cdot Q^b$, where a and b are fitting parameters [*Godsey et al.*, 2009; *Basu et al.*, 2010; *Thompson et al.*, 2011; *Moquet et al.*, 2015; *Moatar et al.*, 2017; *Musolff et al.*, 2017]. A very common metric, relevant also for this study, is based on the value of the *b* exponent, the slope

of the regression in the log(*C*)-log(*Q*) plot, because it is related to the concept of "chemostasis" [*Godsey et al.*,
2009] or "biogeochemical stationarity" [*Basu et al.*, 2010]. A catchment shows "chemostatic" behavior when
despite a sensible variation in discharge, solute concentrations show a negligible variability, i.e., b≅0. Conversely,
positive slopes (i.e., increasing concentrations with increasing discharge) would support an enrichment behavior
when the solute amount grows with discharge and negative slopes (i.e., decreasing concentrations with increasing
discharge) support a dilution behavior with solute mass that does not increase proportionally to the growing
discharge. A solute is typically defined transport-limited if it is characterized by enrichment, while it is called
source-limited in case it dilutes [*Duncan et al.*, 2017a].
The exact mechanisms leading to *C-Q* relations are, to a large extent, an open question, but these relations are
anyway providing insights on solute and/or catchment behavior [*Godsey et al.*, 2009; *Moatar et al.*, 2017]. The
concept of chemostasis emerged in studies that explored the *C-Q* power-law with the aim of demonstrating the
similarities in the export behavior of nutrients [*Basu et al.*, 2010; *Basu et al.*, 2011] and geogenic solutes [*Godsey
et al.*, 2009] across a range of catchments [*Musolff et al.* 2015]. These studies were mostly carried out in
agricultural catchments, where a "legacy storage" was supposed to exist due to antecedent intensive fertilization
practices [*Basu et al.*, 2010; *Basu et al.*, 2011; *Hamilton*, 2012; *Sharpley et al.*, 2013; *van Meter and Basu*, 2015;
*van Meter et al.*, 2016a; *van Meter et al.*, 2016b]. This storage of nutrients might have long-memory effects and it
was considered to buffer the variability of concentrations in streams, leading to the emergence of biogeochemical
stationarity [*Basu et al.*, 2011]. However, biogeochemical stationarity has been questioned outside of agriculturally
impacted catchments [*Thompson et al.*, 2011] and a unifying theory explaining catchment-specific *C-Q* behavior
is not available yet, considering that solutes can show different behaviors in relation to landscape heterogeneity
[*Herndon et al.*, 2015] and to the spatial and temporal scales of measurement [*Gwenzi et al.*, 2017]. Therefore,
approaching the study of solute export and C-Q relations requires the separate analysis of several solutes in as
many catchments as possible with the aim to find, at least, some general behavior that can be characteristic of a
given region or solute. The recent literature is moving toward this direction [*Herndon et al.*, 2015; *Wymore et al.*,
2017] with the aim to sort out the relative influence of climatic forcing, solute properties, and catchment
characteristics on solute behavior in search for generalizations across different catchments.
This study contributes to this line of research investigating a unique dataset of long-term water quality data in
eleven catchments in Switzerland, where multiple solutes were observed at the bi-weekly scale for multiple
decades with limited gaps. We perform the analysis focusing mainly on the temporal domain and by quantifying
magnitude, temporal trends, and seasonality of the in-stream concentrations with the goal of highlighting the long-
term behavior differences across the eleven catchments and investigating the drivers of such differences.
Specifically, we focus on the following research objectives: (i) investigating to which extent the solute
concentrations are influenced by anthropic activities; (ii) exploring the dependence of solute concentrations on
catchment characteristics; (iii) generalizing, if possible, the behaviors of selected solutes across different
catchments by means of the slope in the *C-Q* relations.
## 2. Study sites
Observations used in this study are obtained from the Swiss National River and Survey Program (NADUF[1]), which
represents the Swiss long-term surface water quality monitoring program. This database includes in total 26
monitoring stations located in different catchments. To ensure representativity and robustness of the analysis we
focus only on those stations with at least 10 consecutive years of water quality measurements. This restricts the
database to eleven catchments, the corresponding locations of which are shown in Figure 1. The resulting case
studies include 5 main catchments (Thur - AN, Aare - BR, Rhine – WM, Rhone – PO and Inn - SA), 3 sub-
catchments (Rhone – PO, Rhine – RE and Rhine – DI) and 2 small headwater catchments (Erlenbach and
Lümpenenbach).
Measurements have a temporal resolution of 14 days, which is similar to the resolution of other studies that
analyzed long-term water quality data. In literature, the temporal resolution of water quality observations ranges
namely from weekly [*Duncan et al.*, 2017a; *Duncan et al.*, 2017b; *Gwenzi et al.*, 2017; *Moatar et al.*, 2017;
*Wymore et al.*, 2017] to 14-days [*Hunsaker and Johnson*, 2017] to monthly [*Basu et al.*, 2010; *Thompson et al.*,
2011; *Musolff et al.*, 2015; *Mora et al.*, 2016; *Moatar et al.*, 2017] or even coarser resolution [*Godsey et al.*, 2009].
In fact, only, very rarely higher-frequency databases are collected and thus analyzed (e.g., *Neal et al.*, 2012; *Neal*
*et al.*, 2013; *von Freyberg et al.*, 2017a).
Stream water is analyzed only twice per month, but is collected continuously thus providing samples that represent
a flow-proportional integral of the preceding 14 days. River water is lifted continuously by a submersible pump
into a closed overflow container (25 L) in the station, at a flow rate of 25-75 L min$^{-1}$. From the container, samples
are transferred in 1 mL portions to sampling bottles. The frequency for the transfer of 1 mL samples is proportional
to the discharge monitored continuously by the gauging device in the same station. The discharge-proportional
sampling device is designed to collect 1-3 L of sample per bottle in each period. The sampling mechanism also
allows the simultaneous collection of up to four integrated samples.

---

[1] https://www.bafu.admin.ch/bafu/en/home/topics/water/state/water--monitoring-networks/national-surface-water-quality-monitoring-programme--nawa-/national-river-monitoring-and-survey-programme--naduf-.html

A 14-days sampling frequency is not sufficient for an evaluation of short-term biogeochemical and transport
processes, which might involve solute transformation (e.g., biological processes, in-stream chemical reactions).
These are simply accounted for in a lumped form in the flow-proportional average concentrations collected in a
two-week interval. Conversely, the dataset is especially suitable for the investigation of long-term trends, due to
the length of the time series, which spans from 11 to 42 years (Table 1). Data are collected following ISO/EN
conform methods for water analysis and subsequently validated by means of an extensive quality control as
described in *Zobrist et al.*, 2018. In addition, we inspected the data to take into account possible errors deriving
from fixed detection limits, e.g., deleting the values below the detection thresholds (See Paragraph S1).
The concentrations reported in the database concern the following solute types: (i) geogenic solutes, originating
mainly form rocks weathering, such as calcium ($Ca^{2+}$), magnesium ($Mg^{2+}$), sodium ($Na^+$), silicic acid ($H_4SiO_4$)
and potassium ($K^+$); (ii) deposition derived solutes, as chloride ($Cl^-$); (iii) nitrogen species (nitrate ($NO_3$) and total
nitrogen (TN)); (iv) phosphorus species (dissolved reactive phosphorus (DRP) and total phosphorus (TP)); and (v)
organic carbon species (dissolved organic carbon (DOC) and total organic carbon (TOC)). The time series of these
concentrations are used in the analyses carried out in this study. Furthermore, the dataset includes also the average
discharge, computed as the mean value over the period between two water quality analyses, as well as other
parameters such as water temperature, hardness ($Ca^{2+} + Mg^{2+}$), alkalinity ($H^+$) and pH.
The selected catchments cover most of the Swiss territory. This is characterized by dissimilarities in terms of
morphology, land use, and anthropic pressure, the latter being intended as activities (e.g. fertilization of agricultural
lands, domestic and industrial waste water treatments, industrial sewage disposal into water), which are expected
to have an impact on the river biogeochemistry and to alter the natural background concentrations and their
seasonality. Figure 1 shows the catchments analyzed in this study as identified by the ID reported in Table1.
Catchments are divided into three categories depending on the morphological zone where they are mainly located:
the Swiss Plateau, a lowland region in the north, the mountainous Alpine area in the centre and south, and a third
category that includes catchments spanning both morphologic zones. The choice of this classification criterion is
discussed in the Section 3.1. Geology also differs from one region to another (Figure 1c). The bedrock of northern
Switzerland, the Jura region, is mainly composed of calcareous rocks, while in the Alpine area crystalline silicic
rocks are dominant (Figure 1c). The Swiss Plateau region is instead characterized by the 'Molasse' sedimentary
rocks (Figure 1c), consisting in conglomerates and sandstones of variable composition (e.g. detrital quartz,
feldspars, calcite, dolomite and gypsum) [*Kilchmann et al.*, 2004]. The relative chemical weathering of carbonate
rock and of gypsum are respectively 12 and 40 times higher than the weathering rate of granite or gneiss [*Meybeck*,
1987], thus suggesting that it is a good proxy to consider the Swiss Plateau area as characterized mainly by a
calcareous bedrock (e.g., *Zobrist et al.*, 2018). As the maps in Figure 1d and 1e show, the prevalent land use in the
Swiss Plateau area is agriculture, while the Alpine area is mainly covered by forests and grasslands. Table 1
specifies if the share of agricultural land is cultivated either intensively, i.e. with significant fertilizer applications,
or extensively, e.g. as alpine grasslands, bush land and parks, which are mostly unfertilized [*Zobrist et al.*, 2018].
The main urban centres are concentrated in the northern Switzerland, together with most of the industrial activities,
which represent potential point sources of pollution. The agricultural activities, especially intensive agriculture,
residential and industrial areas are referred in this study as "anthropic pressure", indicating that the sources of
solutes originated from these activities are other than natural. Given the much higher presence of these
anthropogenic factors in the northern Switzerland, the anthropic pressure follows a south-north gradient, although
patches of anthropic pressure are found also within the alpine valleys.
**3. Methods**
**3.1 Magnitude, seasonality and trends**
The magnitude of a solute is evaluated through basic statistics (i.e., median, 25th and 75th percentiles, minimum
and maximum values). These are computed for each solute in each catchment, with the goal of highlighting
differences across catchments, which are the result of catchment heterogeneities and natural and anthropogenic
factors affecting the quantity of a given solute.
The seasonality of discharge and of solute concentrations is analyzed and cross-compared to highlight differences
and similarities of controls that are related to the climatic seasonality and seasonality of man-induced impacts. For
this analysis, catchments are subdivided in the three above mentioned categories: Swiss Plateau, Alpine, and hybrid
catchments (Figure 1). The Swiss Plateau and Alpine catchments have substantially different hydrological regimes
(Figure S1, upper and bottom panels), and represent the main classes of the clusterization proposed by *Weingartner*
*and Aschwanden* (1992). Some of the selected catchments with large draining area include both typologies and are
therefore defined as "hybrid catchments". They are characterised by a seasonality, which is intermediate between
the two end-members (Figure S1, central panel) because the timing of the peak is similar to the one of Alpine
catchments, but the magnitude is less pronounced as in the Swiss Plateau catchments. For this reason, they have
to be treated separately from the other two classes. The hybrid catchments have the highest percentage of lake
surface area in their domains (Table 1), although non-negligible lake fractions are also found in the two other
categories. Large lakes represent a discontinuity in the river network, reducing the fraction of catchment area
directly (without major water mixing effects) contributing to the observed discharge and solute dynamics. The
presence of large lakes contributes to the dampening of the hydro-chemical signal, but its exact quantification is
not straightforward. Aware of the confounding role of large lakes, we apply this classification in order to test if
the seasonality of solutes is related to the seasonality of discharge. With such an analysis we aim at isolating the
effect of the discharge seasonality versus the seasonality of solute concentrations. More specifically, whenever a
solute shows a seasonality different from  the one imposed by climate, we investigate the potential reasons for
such a difference, being it either related to specific catchment characteristics or to anthropic activities.
The comparison between the seasonality of solute and discharge is made through an "index of variability" defined
as the ratio between the mean monthly deviations from the mean of solute concentration and discharge
respectively, where the "deviation" is determined as the average difference between the monthly means and the
annual average value, resulting in the following equation:
$$Index\ of\ variability = \frac{\sum n \left| \frac{Normalised\ mean\ of\ monthly\ deviations\ of\ concentration}{Normalised\ mean\ of\ monthly\ deviations\ of\ discharge} \right|}{n} = \frac{\sum n \left| \frac{\left| \frac{\sum_{i=1}^{12} c_i}{\bar{C}} - 1 \right|}{\left| \frac{\sum_{i=1}^{12} Q_i}{\bar{Q}} - 1 \right|} \right|}{n},$$
where $i$ represents the month of the year, from 1 to 12, and $n$ is the number of the catchments belonging to the
specific catchment class for which the index of variability is computed. In other words, an index of variability
larger the one suggests that the seasonality of the solute is more pronounced than that of discharge, and vice-versa
for an index of variability smaller than one.
Finally, we evaluated the occurrence of trends in the long-term concentration time series at monthly and annual
scale using the monthly average concentration of each solute in each catchment and each year for the entire period.
The statistical significance of trends was tested with the Mann-Kendall test modified to account for the effect of
autocorrelation [*Hamed and Rao,* 1998; *Kendall,* 1975; *Mann,* 1945], fixing a significance level of 0.05. Trends
are investigated and compared across catchments, in order to understand if they are consistent across Switzerland,
thus suggesting the presence of clear drivers underlying the trend, or if they are just occurring in a sub-set of
catchments. The time series span different periods of time, so the results might be impacted by the natural
variability of discharge over the different years. This might be a potential issue, but we observed that in case of
the presence of a trend in discharge (e.g. in the CH catchment, not shown), the patterns of concentrations do not
show any different behavior compared to those observed in other catchments, which our analysis attributes to other
external forcing (e.g., anthropic activities).
**3.2 Concentration-Discharge relations**
The empirical relation between solute concentration and discharge $C = a \cdot Q^b$ was explored separately for each
solute and for each catchment with the objective of investigating solute behaviors across catchments and whether
this behavior can be generalized. The two variables are expected to exhibit in a log-log scale a linear relation,
expressed by mean of the two regression parameters *a*, the intercept with the same dimensions of the concentration,
and *b*, the dimensionless exponent representing the slope of the interpolating line. We focus our attention on the
latter, which determines the behavior of the solute. The Student's t test was applied to verify the statistical
significance of having a *b* exponent different from zero. The level of significance α was set at 0.05. When the p
value was lower than α, the slope identifying the log-linear *C-Q* relation was considered significant and quantified
by *b*, otherwise the slope was considered indistinguishable from zero, thus suggesting no evidence of a dependence
of concentration on discharge.
In each catchment, the time series of discharge were divided into two subsets using the median daily discharge $q_{50}$
to separate flow below the median (low-flows) and flows above the median (high-flows). Hourly discharge time
series were available from the Swiss Federal Office for the Environment (FOEN) at the same river sections and
for the same period of the time series of water quality provided by the NADUF monitoring program. The median
daily discharge was computed from the hourly series, which were aggregated to obtain daily resolution.
Determining the *C-Q* relations separately for high and low-flows allows a finer classification of the solute behavior
into different categories [*Moatar et al.*, 2017], than considering only the dependence on the entire range of
discharge. The three main behaviors – "enrichment or removal" (i.e., positive slope ), "chemostatic" (i.e., near-
zero slope) and "dilution" (i.e., negative slope) – can indeed be the result of mechanisms controlling the runoff
formation and the transport mechanism. Accordingly, we have in total 9 different combinations characterizing the
C-Q relation across high and low flow regimes, which allow assigning distinct behaviors to a given solute.
For solutes that showed long-term trends over the monitoring period, we also investigated the evolution of the *b*
exponent in time. In this case, the concentration and discharge time series were divided into decades and the *C-Q*
relations over all discharge values were computed separately for each decade. The behavioral classification is
performed on a single *b* (i.e., not divided into low- and high-flow *b*), since, differently from the previous analysis
of *C-Q* relations, the focus is on the detection of long-term trends in solute behavior rather than on the
understanding of the processes leading to differences between high and low flows.

## 4. Results

### 4.1 Magnitude

Among the geogenic solutes, $Ca^{2+}$ is the most abundant, most likely due to the composition of the bedrock present
in most of the catchments (calcite, dolomite and anhydrite/gypsum [*Rodriguez-Murillo et al.*, 2014]). In absolute
terms, geogenic solutes and $Cl^-$ have the highest concentrations (≈10-50 mg/L), while phosphorus species
concentrations ($\approx$0.01-0.1 mg/L) are on average one to two order of magnitude less abundant than nitrogen species
($\approx$0.5-1.5 mg/L) and organic carbon ($\approx$1.5-5 mg/L).
Some solutes are constituents of other species, like in the case of nutrients $NO_3$ of TN and DRP of TP. $NO_3$ is
often introduced in catchments as inorganic fertilizer, as DRP, which represents a readily available nutrient for
crops. We computed the ratio between the solute and its component for the two couples ($NO_3$/TN, DRP/TP) and
observed their pattern across the catchments (Figure 2). We take as reference values the ratios in ER catchment,
since, due to limited anthropogenic pressure, it represents the background concentrations of nutrients [*Zobrist*,
2010]. Variations compared to ER values might provide an indication of the ratio of nutrients coming from
anthropic activities. NO3 is the major constituent of TN, since it is about 85% of TN, while DRP contributes much
less to TP, being only its 35%. Both have a decreasing pattern with decreasing catchment anthropogenic
disturbances, although in DRP/TP this pattern is more evident. DRP/TP spans from a maximum of 65% in WM to
a minimum of 22% in ER, while $NO_3$/TN has a maximum of 93% in AN and it is 63% in ER.
Effects of catchment characteristics and human activities on the observed stream solute concentrations can be seen
for certain solutes as shown by Figure 3, where each box shows the measured concentrations in the 11 catchments
and the last box on the right refers to all the catchments grouped together. The catchments, expressed by the
corresponding acronym (see Table 1), are ordered, from left to right, from the most impacted by human activity -
i.e., higher percentage of catchment area used for intensive agriculture - to the least impacted, which is almost
equivalent to considering a south-to-north gradient. The most evident effect of catchment characteristics refers to
the presence of $Ca^{2+}$ and $H_4SiO_4$ in the stream water (Figure 3a). Despite the lower solubility of silicic rocks
compared to the calcareous rocks, $H_4SiO_4$ concentrations in the southern Alpine catchments of Inn (SA), Rhine
(DI) and Rhone (PO) are significantly higher than the median value across catchments. The impact of human
activities, instead, is more evident in $Na^+$ and $Cl^-$ concentrations. These are showing, basically, the same pattern
across catchments (Figure 3b), indicating that they are most likely influenced by the same driver, which is the
spreading of salt on roads during winter months for deicing purposes. We consider the spreading of deicing salt
an anthropic activity related to the presence of inhabitants in a catchment. DOC and TOC concentrations are very
high in Lümpenenbach (LU) and Erlenbach (ER) catchments (Figure 3c), which are the smallest catchments with
the highest average yearly precipitation rate and very low anthropic presence. Thur (AN) and Aare (BR)
catchments also show DOC and TOC concentrations higher than the average, but in these catchments the presence
of wastewater treatment plants can influence TOC concentrations. Finally, nutrients, such as nitrogen species and
phosphorus species, which are connected with anthropic activities (fertilization, wastewater treatment plants) show
a relatively clear decreasing median concentrations from the most to the least impacted catchment (Figure 3d).
Indeed, regressing median solute concentration with the percentage of intensive agricultural land and the
inhabitants density (Table S1a), gives a statistically significant dependence for some nutrients (i.e., $NO_3$, TN,
DRP). Because the catchments that are mostly impacted by agricultural activities are mainly located in the Swiss
Plateau, a significant positive correlation between nutrients and the percentage of Swiss Plateau area of the
catchment exists; conversely, we observe a significant negative correlation with the percentage of the Alpine area.
One should note, however, that the correlation is performed on 11 catchments only, so that lack of significance
should be interpreted with care. Indeed, if we extend the correlation analysis to the *b* exponent derived from the
*C-Q* relations analysis – thus implicitly accounting for the complex interactions between catchment
geomorphology, land use, hydrological processes and solute releases – with the same catchment characteristics
(e.g., *Moatar et al.*, 2017) the correlation becomes weaker and, basically, not significant for any solute (Table S1b
and Table S1c).
**4.2 Seasonality**
Different climates and catchment topographies determine various hydrological responses, as we can observe in
Figure S1 from the analysis of discharge seasonality across the eleven catchments, expressed through the monthly
average streamflow normalized by its long-term average. We present the results with the catchments divided in 3
groups as previously explained. The partition into these classes helps in highlighting the effects of topography,
climatic gradient and somehow also the impact of anthropic activities since it follows a similar south to north
gradient. The seasonality of streamflow in Swiss Plateau catchments is determined by a combination of
precipitation and snowmelt.  The peak flow is typically observed in spring and is not much higher than the average
in the other months. Alpine catchments, instead, show stronger seasonality induced by snow and ice-melt in spring
and summer, which generates higher streamflows than in the other months. Hybrid catchments exhibit flow peaks
in June-August similarly to the Alpine ones, but the deviation from the average value is less pronounced.
The deviations of discharge and concentration are compared using the index of variability (Section 3.1) for each
morphological class of catchments (Figure 4). Only few solutes show a value of the index higher than 1. This
indicates that seasonality of solute concentrations is generally lower or much lower than the seasonality of
streamflow. This is especially true for the Alpine catchments, where the marked seasonality of streamflow seems
to dominate the variability of concentrations. For TP this index is higher than one in Alpine catchments, and also
the highest compared to the other two typologies. In Swiss Plateau and hybrid catchments, instead, only solutes
impacted by human activity ($Na^+$, $Cl^-$, nitrogen species and DRP) show a ratio close or even higher than 1.
DOC and TOC concentrations are characterized by low indexes of variability, especially in the hybrid catchments.
The patterns of the index of variability across different morphologies can be classified into three categories,
represented by the symbols A, B and C in Figure 4. The monotonic line in A type refers to those solutes, the
variability index of which changes across morphologies solely as a result of the seasonality of streamflow ($Ca^{2+}$,
$Na^{2+}$, $K^+$ and $Cl^-$). Type B solute ($Mg^{2+}$, TP, DOC and TOC) response shows a higher variability index in Alpine
catchments compared to types A and C, thus indicating that, among the factors controlling the seasonality of
biogeochemical response, there are factors that are specific to the Alpine environment, which are discussed in
Section 5.2. The type C pattern, instead, refers to solutes related to fertilization ($NO_3$, TN and DRP) and to $H_4SiO_4$,
which is a product of weathering and only minimally involved in biological processes. These solutes are
characterized by a much lower variability index in Alpine catchments than in hybrid and Swiss Plateau catchments.
Difference in their regime are further discussed in Section 4.
The analyzed solutes show different intra-annual dynamics. For instance, despite the quite pronounced streamflow
seasonality of the Rhine River at Rekingen (hybrid catchment used as a representative example), solute
concentration patterns shows different seasonal cycles (Figure S2). $Ca^{2+}$, $Mg^{2+}$, $Na^+$, $K^+$, $Cl^-$, $NO_3$ and TN
concentrations peak in February-March and have lower values during spring-summer period, showing a pattern
opposite to that of streamflow. $H_4SiO_4$, instead, has a shifted seasonality compared to the other solutes, peaking in
December-January. Phosphorus species together with organic carbon species do not show any consistent
seasonality over the year.

## 4.3 Trends

Long-term trends in the concentration time series are investigated with respect to the seasonal cycle for each year
separately (Figure S2). One catchment (Rhine-Rekingen) is taken as an example for illustration purposes but
generality of trend results is discussed in the following.
Focusing on the long-term horizon, different dynamics can be observed across various solutes. Some of them show
visible trends: for instance $Cl^-$ has increased from 1970s to 2015, while phosphorus species have decreased
considerably. Some solutes have different trends across different catchments. A generalization of long-term
patterns is shown in Figure 5 for the three main detected behaviors. The upper panel represents the occurrence of
an evident trend, either increasing (as in the example of $Cl^-$) or decreasing (e.g., TP). $Na^+$, $Cl^-$, DRP and TP belong
to this category. While $Na^+$, $Cl^-$ have increased in time, DRP and TP have decreased in the monitoring period, as
the monthly trends in Table S1a show (see Figure 6 for DRP only).
The middle panel shows a non-monotonic trend. This is typical of $Mg^{2+}$, which first increased in most catchments
(1970s-1990s) and then decreased (1990s-2015). $K^+$, TN and TOC also show this type of trend in most catchments.
Finally, the lower panel of Figure 5 shows a number of solutes ($Ca^{2+}$, $H_4SiO_4$, $NO_3$ and DOC) that do not exhibit
any long-term trend, although analysis on a monthly base revealed some significant trends (Table S1c).

**4.4 C-Q relations**

Concentration-discharge relations were computed for all the solutes across all the catchments as summarized in
Table 2. For each solute, we computed the number of catchments showing a given specific behavior, which we
denoted with the combination of the symbols "+" (i.e. enrichment/removal), "-" (i.e. dilution) and "=" (i.e.
chemostatic behavior) for discharge above and below the median.
Geogenic solutes are mostly characterized by dilution. The only exception is $H_4SiO_4$, which shows 6 different
behaviors across the 11 catchments, making impossible to identify the most representative behavior for this solute.
This is the case also of other species (nitrogen species, TP and organic carbon species), which show at least three
different behaviors across catchments. Silicium is mainly generated through rock weathering, but it is also involved
in biological processes, which might influence its behavior across catchments.
Overall, dilution is dominant for all solutes in both low- and high-flow conditions, as it occurs respectively in 65%
and 57% of the catchments. Therefore, even in low-flow conditions, the solute transport is mainly source limited
across catchments. Only sediment-related solutes (i.e., TP, TOC), show a marked transport limited behavior. The
label "sediment-related solutes" comes from the fact that phosphorus and organic carbon are bonded to soil
particles and, when soil is eroded, carbon- and phosphorus-rich soil particles are mobilized by flowing water. In
such conditions, soil erosion becomes one of the main contributor to the phosphorus and organic carbon load into
the rivers. We investigated also C-Q relations for suspended sediment concentrations and they show increasing
slope across all the catchments, indicating, as expected, higher erosion rates in presence of high flow conditions.
Only 29% of the catchment-solute combinations have different behaviors between low- and high-flow conditions
and therefore the C-Q relations are represented by bended lines, having different slopes between low- and high-
flow conditions.
$NO_3$ and DOC represent a conspicuous component of TN and TOC respectively, but $NO_3$ shows almost the same
behaviors of TN, in spite of a different distribution across catchments, while DOC and TOC behave differently.
Phosphorus species also show different behaviors, consistently with the fact that DRP represents only a small
fraction of TP.
Since in the trend analysis we identified four species ($Na^+$, $Cl^-$, DRP and TP) that are characterized by remarkable
long-term trends, we investigated if such a significant change in magnitude has an effect on the C-Q relation
analyzing the temporal changes of the b exponent. The changes in the value of b across all catchments with record

length longer than 30 years during different decades is shown in the left panel of Figure 8, whereas the right panel of Figure 8 shows an example of variation of the TP C-Q relations across decades for the human-impacted catchment of Aare – BR and the Alpine catchment of Rhone - PO. Although the observed concentrations of all four solutes - $Na^+$, $Cl^-$, DRP and TP - are characterized by the presence of evident trends in time, the behaviors in the C-Q relation differ. $Na^+$ and $Cl^-$ have a constant b exponent across decades, while phosphorous species show increasing b, which, in some catchments, leads to a switch from a behavior of dilution to one of enrichment.

## 5 Discussion

### 5.1 Influences of human activities on solute concentrations

The cause-effect relation between the observed in-stream concentrations and the anthropic activities is sometimes evident in the concentration magnitude, seasonality, and long-term trends. Phosphorus and nitrogen are the main nutrients applied for agricultural fertilization and, a decreasing pattern of their magnitude from mostly intensive agricultural catchments to forested catchments is observed (Figure 3d). Indeed, taking the concentrations of $NO_3$ and DRP registered at ER as reference background of natural concentrations [*Zobrist*, 2010], corresponding to 0.20 mg/L of $NO_3$, 0.38 mg/L of TN, 0.002 mg/L of DRP and 0.02 mg/L of TP, the concentrations in all the other catchments are significantly higher. For example, the most impacted AN catchment recorded median concentrations of 2.50 mg/L of $NO_3$, 3.03 mg/L of TN, 0.06 mg/L of DRP and 0.15 mg/L of TP. Following the stoichiometric composition of plants, nitrogen species concentrations are one order of magnitude higher than phosphorus species concentrations (Figure 3d). Nitrogen is the main nutrient required for crop growth [*Addiscott*, 2005; *Bothe*, 2007, *Galloway et al.*, 2004; *Zhang*, 2017] and indeed $NO_3$ is one of the main components of fertilizers applied in agriculture. $NO_3$ represents a large fraction of TN (Figure 2). The variability of the ratio between average $NO_3$ and TN concentrations across the different catchments, is comparable with that estimated by *Zobrist and Reichert* (2006), who observed a variation from 55% in Alpine rivers to 90% for rivers in the Swiss Plateau. Both $NO_3$ to TN and DRP to TP ratios show a decreasing trend from more to less anthropic-impacted catchments, the range of variability being, however, higher for phosphorus species (from about 0.6 in Thur river to about 0.2 in Inn River). The DRP/TP ratios across catchments can be explained as the result of the cumulative effect of two main factors: the lower DRP input due to less intensive agricultural activity in the Alpine zone and the higher share of phosphorus sourced by suspended sediments contributing to TP in Alpine catchments due to generally higher erosion rates.

Anthropic activities affect also the seasonality of certain solutes. In Figure 4, we assigned the pattern "C" to those solutes (i.e., $H_4SiO_4$, $NO_3$, TN and DRP) characterized by a much lower index of variability in Alpine catchments

than in hybrid and Swiss Plateau catchments. For those solute concentrations, variability in Swiss Plateau and
hybrid catchments are comparable or higher than streamflow variability, while in Alpine catchments streamflow
seasonality is much stronger than solute seasonality. A non-negligible fraction of these solutes is introduced
through agricultural practices or by means of other human activities. Their input is characterized by its own
seasonality, which influences the solute dynamics and makes it comparable or larger than the discharge seasonality,
a behavior non-observable for most geogenic solutes (Figure 4). An additional evidence supporting this result is
represented by the patterns of the average monthly discharge and solute load (computed as the product between
concentration and discharge) normalized by the respective average value. This representation is made for $Ca^{2+}$,
originated by rocks weathering, and $NO_3$, mainly of anthropic origin (Figures S3a and S3b). The plot, inspired by
the analysis of *Hari and Zobrist* (2003), shows how the seasonality of $Ca^{2+}$ load follows well the seasonality of
discharge across all catchments, while $NO_3$ load has its own seasonality in the catchments with the largest
agriculture extent, especially in the first part of the year. Indeed, in the case of $NO_3$, there is no correspondence
between the seasonality of discharge and load (e.g. the time of maximum discharge does not coincide with the
time of maximum or minimum load), thus suggesting that the input is characterized by an independent seasonality.
Anthropic activities do not only influence the average solute concentrations and the seasonality, but also the long-
term dynamics. $Na^+$ and $Cl^-$ show clear positive trend in time (Table S1a), largely because of the increasing
application of deicing salt (NaCl) [*Gianini et al.*, 2012; *Novotny et al.*, 2008; *Zobrist and Reichert*, 2006]. A clue
of the cause-effect relation between deicing salt application and increased $Na^+$ and $Cl^-$ concentrations in stream
water comes from stoichiometry. The molar ratio between $Na^+$ and $Cl^-$ in salt is 1:1, therefore, the closer to 1 is
the ratio computed on observed in-stream concentrations, the more likely deicing salt may be the driver. Figure S4
shows the boxplot of the Na:Cl molar ratio across catchments and it is clear that catchments with higher population
density show values closer to one. However, the Erlenbach (ER) and Lümpenenbach (LU) catchments, which do
not show any increasing long-term trend neither in $Na^+$ nor in $Cl^-$ concentrations, show Na:Cl values higher than
one, consistently with catchments with the low population density (i.e., Rhone (PO), Rhine (DI) and Inn (SA)). In
this respect, *Müller and Gächter* (2011) analyzed the phenomenon of increasing $Cl^-$ concentrations in Lake Geneva
basing their analysis on the NADUF data at the Rhine-Diepoldsau (DI) station. The concentrations detected by the
water quality monitoring station are much lower than the amount of the input of salt declared by the cantonal
authorities and the increasing trend characterizes the whole year and not only the winter months. These two factors
suggest that an accumulation effect with a long-memory in the system might exist. The salt could be stored
somewhere in the soil or in the groundwater and could be progressively delivered to the streams over years.
However, this is difficult to assert conclusively since the salt input is uncertain. Indeed, estimating the input of salt
used for deicing purposes is not trivial, due to the lack of reliable data [*Müller and Gächter*, 2011]. Official sources
[*EAWAG*, 2011] state that improved technologies have enabled a sensible decrease of the specific amount of spread
salt (from 40 g/m$^2$ in 1960s to 10-15 g/m$^2$ of today), but the total amount of salt still shows increasing trend, likely
because it is spread more often and on wider surfaces. The recent study of *Zobrist et al.* (2018) uses as a proxy for
salt consumption the salt production by Swiss salt refineries, and claims an increase from 360 Gg NaCl year$^{-1}$ in
the 1980s to 560 Gg NaCl year$^{-1}$ to the present, thus supporting the observed positive trend.
A positive cause-effect relation between anthropic activity and solute concentration in terms of trend is also shown
for phosphorus species, which decreased consistently since 1986 (Figure 6), when the phosphate ban in laundry
detergents was introduced in Switzerland [*Jakob et al.*, 2002; *Rodriguez-Murillo et al.*, 2014; *Prasuhn and Sieber*,
2005; *Zobrist and Reichert*, 2006; *Zobrist*, 2010].
A non-monotonic trend emerged from the analysis of long-term data for Mg$^{2+}$, K$^+$, TN and TOC (Figure 5).
Considering for example Mg$^{2+}$, *Zobrist* (2010) focuses the trend analysis over the period 1975-1996 on Alpine
catchments and observes a similar non-monotonic increasing-decreasing pattern. *Zobrist* (2010) attributes this
pattern to an increase of water temperature, which is evident for the Rhine and Rhone rivers. For Rhine and Rhone
rivers, our results support the conclusion of *Zobrist* (2010) because at the decreasing-increasing trend of Mg$^{2+}$
corresponds a reverse increasing-decreasing trend in Ca$^{2+}$. This is consistent with the temperature dependence in
calcite solubility. However, in the Thur catchment (AN and HA catchments) which is mainly agricultural, the non-
monotonic trend of Mg$^{2+}$, does not correspond to a trend in Ca$^{2+}$. Since Mg$^{2+}$ can cumulate through fertilizer
applications and carbonates weathering (i.e., Mg$^{2+}$ production) can be affected by N-fertilizers and manure
application [*Hamilton et al.*, 2007, *Brunet et al.*, 2011], we hypothesize that fertilizers might also have an impact
on the Mg$^{2+}$ long-term dynamic. In this respect, the analysis of monthly trends of Mg$^{2+}$ (Table S1b) shows a more
evident increasing trend for agricultural than for non-agricultural catchments. For K$^+$ the difference across the
gradient of agricultural pressure is not as remarkable as for Mg$^{2+}$. Monthly trends of TN and DOC revealed
increasing tendency in the first months of the year (January-April) and decreasing ones in the last part of the year
(August-December), thus suggesting that they are induced either by streamflow trends (*Birsan et al.*, 2005) or by
biogeochemical processes, which have a pronounced seasonality related to temperature and moisture controls
rather than to human activities.
In summary, the anthropogenic signature is clearly detectable in the water quality of catchments with an important
fraction of intensive agriculture and relatively high population density, especially in the magnitude of
concentrations of nutrients (i.e., nitrogen and phosphorous species), in the increasing long-term trends of Na$^+$ and
Cl$^-$ and, a positive outcome of environmental regulations, in the decreasing long-term trends of phosphorous
species. Moreover, the seasonality of nutrients differs considerably from the seasonality of naturally originated
solutes (e.g., geogenic solutes).

**5.2 Influence of catchment characteristics on magnitude and trends of solute concentrations**

A statistically robust link between catchment characteristics and river biogeochemical signatures is not
straightforward, because the spatial heterogeneity in river catchments and the limited sample size, make the search
for cause-effect relations between catchment characteristics and in-stream concentrations challenging. However,
catchment characteristics play a role for certain solutes and we found evidence of their impact especially in the
magnitude and seasonality of solute concentrations. First, the geological composition of the bedrock influences
the weathering products, increasing $Ca^{2+}$ concentrations in mostly calcareous catchments (northern Switzerland)
and of $H_4SiO_4$ in silicic catchments (Alpine catchments in central and southern Switzerland). The catchments DI,
PO and SA, which are entirely located in the Alpine area (Table 1) and mainly lay on crystalline bedrock (Figure
1c), have higher concentration of silicic acid (Figure 3a) along with a lower concentration of $Ca^{2+}$ in comparison
to the other catchments, being the AN in the Swiss Plateau area (Table 1) an exception, which is characterized by
a concentration of silicic acid that is comparable to that of Alpine catchments. The influence of lithology was
identified before in literature, with, for instance, high $Ca^{2+}$ concentrations in one of the tributaries of the Amazon
River attributed to the presence of carbonate-richer lithology in the corresponding catchment [*Baronas et al.*, 2017;
*Rue et al.*, 2017; *Torres et al.*, 2017].
In the seasonality analysis, the classification of catchments into classes helps highlighting the impact of the
topography on the solute variability. In the Alpine catchments, discharge seasonality generally dominates the
seasonality of solute concentrations, except for TP, which is related to the presence of suspended sediments in the
streamflow caused by higher erosion rates [*Haggard and Sharpley*, 2007]. Indeed, suspended sediment
concentrations, coming from erosion, are much higher in Alpine catchments, excluding the two small headwater
catchments LU and ER, than in the others (Figure S5). Furthermore, erosion represents a source also for DOC and
TOC [*Schlesinger and Melack*, 1981]. TP, DOC and TOC together with $Mg^{2+}$ have been classified as solutes
belonging to "B" class (Figure 4), i.e. their concentration patterns show higher variability in Alpine catchments
than across other classes. The driver of $Mg^{2+}$ variability is, however, less clear than for the others. The higher
variability of its concentrations in Alpine catchments in comparison to other catchments might be due to the
presence of glaciers. Rhone, Rhine and Inn rivers include considerable glaciated areas in their catchments and this
might have an effect on magnesium concentration in stream water. The chemistry of glacier water is generally
characterized by low water-rock contact times because the volume of water and the flow rate are high so that the
time water molecules interact with sediments is relatively short [*Wimpenny et al.*, 2010]. Therefore, water sourced
by glacier melt can have a dilution effect in terms of $Mg^{2+}$ and this explains why $Mg^{2+}$ concentrations are
significantly higher during low-flow periods than during high-flow periods. This is also consistent with the
observations of other studies, e.g. *Ward et al.* (1998), *Wimpenny et al.* (2010a), *Wimpenny et al.* (2010b).
Weathering processes in Alpine environments are also studied using isotope data (e.g. *Tipper et al.* (2012), *von*
*Strandmann et al.* (2008)). These results underlay the uncertainty on the processes determining weathering
products as $Mg^{2+}$. Besides the contribution of glacier-sourced water to streamflow and biological processes
affecting $Mg^{2+}$ concentrations [*Wimpenny et al.*, 2010b], dissolution of bedrock non-proportional to its
composition [*Kober et al.*, 2007], which is likely to take place in presence of carbonate-poor glacial sediments
[*McGillen and Fairchild*, 2005], might also play a role. Carbonate rocks might dissolve with preferential release
of $Mg^{2+}$, which therefore contributes strongly to solute fluxes in rivers. This phenomenon has been observed also
in the Swiss Alps (Haut Glacier d'Arolla), where carbonate contents of sediments are of the order of 1% [*Brown*
*et al.*, 1996; *Fairchild et al.*, 1999], but their contribution to solute fluxes is much higher [*McGillen and Fairchild*,
2005].
Catchment size or precipitation might also influence river solute concentrations. This is evident from the behavior
of the Lümpenenbach (LU) and Erlenbach (ER) catchments, which are three orders of magnitude smaller than the
other catchments considered in the study and show median concentrations lower than those of the other catchments.
This is true for all solutes, except DOC and TOC, the concentrations of which are the highest in Erlenbach (ER)
and Lümpenenbach (LU) rivers. These catchments are situated in Alptal valley, which is characterized by more
humid climate (double annual precipitation), compared to other catchments. Recently, V*on Freyberg et al.* (2017b)
analyzed isotope data of 22 catchments across Switzerland, including LU and ER, computed the young water
fraction (i.e., the proportion of catchment outflow younger than approximately 2-3 months) across 22 Swiss
catchments and tested its correlation with a wide range of landscape and hydro-climatic indices. They inferred that
hydrological transport in LU and ER is dominated by fast runoff flow paths, given the humid conditions and low
storage capacity when compared to other catchments. DOC exports have typically been associated with near-
surface hydrologic flow paths [*Boyer et al.*, 1997, *Tunaley et al.,* 2016; *Zimmer and McGlynn*, 2018], thus offering
a possible explanation for the higher concentration of DOC and TOC in these catchments.
In summary, the comparison among catchments highlighted differences in magnitude of silicic acid and calcium,
likely due to the different underling lithology. Steeper morphologies show higher sediment transport in surface
water, which is consistent with the observation of pronounced seasonality of sediment-binding solutes (i.e., TOC
and TP) in the Alpine catchments. The headwater catchments ER and LU, which are smaller and wetter than the
other case studies, show a peculiar behavior with enhanced DOC and TOC concentrations, likely as a consequence
of humid conditions, near surface and/or surface flow, and low storage capacity.
**5.3 Consistency of solute behaviors across catchments**
This study showed that concentration-discharge relations reveal nearly chemostatic behavior for most of the
considered solutes across catchments, i.e. analyzed solute concentrations vary a few order of magnitude less than
discharge (Figure S6). This outcome agrees with other studies (e.g., *Godsey et al.*, 2009; *Diamond and Cohen*,
2017; *Kim et al.*, 2017; *McIntosh et al.*, 2017). We found that the in-stream biogeochemical signal is highly
dampened, coherently with other studies [*Kirchner et al.*, 2000; *Kirchner and Neal*, 2013], but different behaviors
of solutes could be nonetheless detected in the log(C)-log(Q) space, thus allowing a partition into four categories,
as suggested by *Moatar et al.* (2017). A representation of such partitioning is offered in Figure 7, where the space
between the negative-slope line and the near-horizontal line represents the dilution behavior, and the space
delimited by the positive-slope line and the near-horizontal line represents the enrichment or removal behavior. In
fact for low-flow conditions (i.e. $q<q_{50}$) this is typically associated with biogeochemical processes of solute
removal (e.g., nitrification), while for high-flow conditions (i.e. $q>q_{50}$) it is generally associated with the capacity
of the flow to entrain particles containing the solute. Such a description provides a different point of view of *C-Q*
relations compared to the existing literature since the subdivision between low- and high-flow conditions allows a
more detailed investigation of the processes potentially determining the observed solute behaviors. However, the
14-days frequency sampling does not allow a direct detection of short-scale processes and especially fast flood-
waves. This limitation could contribute to the low percentage, only 29%, of cases in which a solute switches the
behavior between low-flow and high-flow conditions. Additional uncertainty is due to the choice of the median
daily discharge as breaking point for the curves. However, in a recent study, *Diamond and Cohen* (2017) tested
various breaking points for the *C-Q* relations of different solutes with most of the breaking points centered on
approximately the median flow supporting our choice. In search for generalizations, we assigned a solute to each
specific class if the same behavior was observed in at least 60% of the analyzed catchments. Geogenic solutes are
grouped in a single circle since almost all of them show a dilution behavior. Only $H_4SiO_4$ does not show a clear
signal, probably because, although to a minor extent, it is involved in complex dynamics related to biological
processes [*Tubaña and Heckman*, 2015], which can affect its behavior. The diluting behavior of geogenic solutes
is a quite well consolidated fact in the literature [*Godsey et al.*, 2009; *Thompson et al.*, 2011; *Baronas et al.*, 2017;
*Diamond and Cohen*, 2017; *Hunsaker and Johnson*, 2017; *Kim et al.*, 2017; *Moatar et al.*, 2017; *Winnick et al.*,
2017; *Wymore et al.*, 2017] and this study contributes to this body of knowledge confirming this behavior.
Residence time is a fundamental hydrological variable for weathering products, since it is related to the weathering
rates and therefore to the resulting solute concentration [*Maher*, 2010]. Catchments that show chemostatic behavior
(e.g., BR for $Ca^{2+}$ or WM for $H_4SiO_4$) likely have average water residence times that exceed the time required to
reach chemical equilibrium, while a dilution behavior is expected when residence times are generally shorter than
required to approach chemical equilibrium [*Maher,* 2011]. Our results suggest that the concentrations of geogenic
solutes across the catchments are far from the equilibrium, which is likely due to relatively fast hydrological
response of Alpine and sub-alpine catchments also associated with substantial precipitation amounts. However,
very likely the residence time and the flow pathways are highly heterogeneous in Alpine catchments with water
from different sources having different biogeochemical characteristics [*Torres et al.*, 2017 and *Baronas et al.*,
2018]. Therefore, flow paths with sufficiently long residence time for reaching chemical equilibration must exist
but they do not leave a major signature on the examined geogenic solutes. In conclusion, there is a quite high
confidence in claiming that geogenic solutes are characterized by a dilution behavior.
The $Cl^-$ solute is also clearly characterized by dilution and our results are in agreement with other studies
[*Thompson et al.*, 2011; *Hoagland et al.*, 2017; *Hunsaker and Johnson*, 2017].
$NO_3$ relations with discharge are less clear [*Aguilera and Melack*, 2018; *Butturini et al.*, 2008; *Diamond and*
*Cohen*, 2017; *Hunsaker and Johnson*, 2017], but this study highlighted a dilution behavior also for $NO_3$ in the
majority of catchments for both low-flow and high-flow conditions. This result partially agrees with the
observations of *Wymore et al.* (2017), who claimed that $NO_3$ shows variable responses to increasing discharge. In
fact, we observed that while dilution is evident in 80% of the catchments for low-flow conditions, this percentage
drops to 63% for high-flow conditions. Although $NO_3$ is one of the main components of TN (Figure 2), TN does
not show the same behavior. For low-flows, TN is also characterized by dilution, but for high-flows TN shows
chemostatic behavior in about 70% of catchments.
The behavior of phosphorus and its compounds is neither clear. For low-flows, DRP behaves chemostatically in
about 40% of catchments, but dilutes in about 60% of catchments. TP behavior could not be classified due to its
variability across catchments for low-flows, whereas, for high-flows, it clearly shows hydrological export in 90%
of catchments, because of increased suspended sediments concentration. In-stream sediments can be, however,
both source and sink for phosphorus [*Haggard and Sharpley*, 2007], as high suspended sediment concentrations
in rivers favor the sorption of phosphorus to particles thus lowering DRP concentrations [*Zobrist et al.*, 2010]. For
high-flow conditions, we observed various DRP behaviors across catchments (about 45% of dilution, 45%
chemostatic and 10% enrichment), so that a clear classification is not possible. The weak correlation between DRP
and suspended sediments concentration suggests that the sorption of phosphorus to particles is not the only and
most influencing factor of DRP dynamic.
TOC is the only solute characterized by enrichment in both low-flow and high-flow conditions. DOC was proved
by a set of studies to exhibit an enrichment behavior (e.g., *Boyer et al.*, 1996; *Boyer et al.*, 1997; *Butturini et al.*,
2008; *Hornberger et al.*, 1994; *McGlynn and McDonnell*, 2003; *Perdrial et al.*, 2014; *Wymore et al.* (2017)), but
our results are in this respect highly uncertain for low-flows and suggest a chemostatic behavior for high-flows.
*Wymore et al.* (2017), for instance, analyzed the biogeochemical response in the Luquillo catchment in Puerto
Rico and detected an enrichment behavior. This catchment is mainly covered by the tropical forest and
characterized by very wet conditions ($\approx$ 4500 mm/yr or rainfall). This is the likely reason leading to higher DOC
concentration with increasing streamflow. The underlying mechanism could be that of a larger share of streamflow
coming in wet conditions from shallower soil pathways [*von Freyberg et al.*, 2017b], which are generally organic-
richer than the deeper horizons hosting lower DOC quantities [*Evans et al.*, 2005]. Our study seems to confirm
this hypothesis, as the wettest catchments analyzed in this study (Erlenbach (ER) and Lümpenenbach (LU)) show
enrichment of DOC at least for low-flow conditions. These are likely mainly dominated by sub-surface flow, thus
confirming the impact of soil wetness in the unsaturated zone on DOC behavior for undisturbed catchments
characterized by wet conditions.
The results of this study also showed that the variability of solute magnitude in the long-term can play a role in the
definition of the solute behavior. $Na^+$ and $Cl^-$ show dilution during the entire monitoring period, despite the
increasing concentrations through time (Figure 8). However, DRP and TP switch from highly negative b exponent
of the C-Q power-law relation to even positive b (Figure 8), after the time when the measures to reduce the
phosphate input were introduced (Figure 6). Such measures [*Zobrist and Reichert*, 2006] lead to a conspicuous
decrease of DRP concentration and partially also of TP. Therefore, the fraction of DRP in TP decreased in time
(Figure S7) and the other TP components became more important than DRP in the definition of TP behavior.
Among these, the component carried with sediments might be responsible for the switch, which took place in all
the analyzed catchments, from dilution to enrichment across the last four decades. DRP also shows increasing
trend of the b exponent of the C-Q relations across decades, but only in two catchments (AN, WM) the behavior
switches from dilution to enrichment. This means that when DRP inputs were higher, the transport was not source
limited, while decreasing the input forced DRP to have a more chemostatic behavior, probably because the input
became so low that the phosphorus transport is controlled by a legacy of phosphorus stored in the soil, which was
accumulated during the years of undisciplined agricultural practices [*Sharpley et al.*, 2013; *Powers et al.*, 2016;
*van Meter et al.*, 2016a].

## 6 Conclusions

The long-term water quality data analysis of this study was designed for understanding the signature of catchment characteristics and the influence of anthropic activities on solutes concentrations observed in Swiss rivers. The analysis of magnitude, seasonality, and temporal trends revealed clear cause-effect relation between human activities and certain solute concentrations (i.e., $Na^+$, $Cl^-$, $NO_3$, DRP). Indeed, changes in the anthropic forcing (e.g., phosphate ban or increased deicing salt) overwhelm the natural climatic variability and are clearly reflected by changes in magnitude of solutes like DRP, TP, $Na^+$, $Cl^-$. The seasonality of anthropogenic-related solutes (i.e., $NO_3$, TN, DRP and TP) in the catchments in the Swiss Plateau more impacted by human activities is clearly altered compared to the seasonality of Alpine catchments.

The detection of the signature of catchment characteristics is less straightforward and can be only captured in a quantitative but not statistically significant way due to the spatial heterogeneity of catchment characteristics and the relatively small sample size (11 catchments). Although the solute export is the result of multiple complex processes, catchment topography, geology and size are expected to have a role in determining solute concentrations, especially of weathering solutes, whose concentrations are influenced by the bedrock composition, and sediment-binding substances (i.e., TP, TOC and DOC) which have an enrichment behavior in catchments characterized by steeper morphologies and higher erosion rates. While we see evidence for a role of catchment characteristics, these influences are relatively minor in our analysis.

The analysis of the empirical *C-Q* power-laws was used to investigate and possibly obtain a generalizable classification of solute behaviors. Repeating the analysis for low-flow and high-flow conditions provides a more detailed description of solute behaviors, in comparison to most of the previous literature. The variability of solute concentration is generally much smaller than that of streamflow, which, in first instance, would support a chemostatic behavior. However, the overall dominant behavior across solutes and catchments is dilution. For many solutes, this result is consistent with other studies (i.e., geogenic solutes and $Cl^-$). Sediment-binding substances (TP, DOC and TOC) show, however, an enrichment during high-flow events, while for other solutes it is not possible to define a clear behavior (e.g., DRP).

Finally, we observed that anthropic activities affect not only the magnitude of concentrations of solutes in rivers, but also their seasonality and long-term dynamics. Remarkable variation in long-term dynamics, moreover, might also determine changes of solutes behavior in time, as we demonstrated for DRP and TP. This time-varying perspective of solute behaviors represents a novelty in literature and gives a clear quantitative evidence that

anthropic activities might influence also the *C*-*Q* relations. Together with the small sample size, one of the main limitations of the study is the coarse temporal resolution of the water quality data that prevents the direct analysis of (solute) fast response times associated with flood dynamics. Luckily, the advancement of technologies in high-resolution concentration measurements research [*von Freyberg et al.*, 2017a] will alleviate this limitation in the future. Despite the above limitations, the above results reinforce and extend the current knowledge on the biogeochemical responses of rivers, demonstrating that long-term observations allow identifying various aspects of anthropic activities on the solute inputs to rivers.

## Acknowledgments

We acknowledge Ursula Schoenenberger for providing the database used for making Figure 1 and Stephan Hug for the information about the NADUF program. River discharge and water quality data were kindly provided by the Swiss River Survey Programme (NADUF; http://www.naduf.ch). We aknowledge Marius Floriancic for providing the macro-geology classes map and for the fruitful scientific discussion. This study was supported by the DAFNE Project (https://dafne.ethz.ch/), funded by the Horizon 2020 programme WATER 2015 of the European Union, GA no. 690268.

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

**List of Tables**
*Table 1:* Description of the catchments. The selected catchments are characterized by different size, elevation and average yearly precipitation. Four catchments are entirely
Alpine (ER, PO, DI, SA), while the others encompass different morphologies (Swiss Plateau and pre-Alpine areas). The data are sourced by the catchment descriptions included
in the NADUF database

| Catchment | Catchment ID | Area (km$^2$) | Average elevation (m a.s.l.) | Mean annual precipitation (mm/y) | Mean annual discharge (mm/y) | Lake area (%) | Morphology | | | Agriculture | | Inhabitants density (inhab/km$^2$) | Period of available data | Number of consecutive years |
|---|---|---|---|---|---|---|---|---|---|---|---|---|---|---|
| | | | | | | | Swiss Plateau (%) | Alps (%) | Other (%) | Int. (%) | Ext. (%) | | | |
| Thur – Andelfingen | AN | 1'696 | 770 | 1'429 | 880 | 0.1 | 50 | 23 | 20 | 51.9 | 10.6 | 222.9 | 1981-2015 | 35 |
| Aare –Brugg | BR | 11'726 | 1'010 | 1'352 | 847 | 3.6 | 38 | 23 | 30 | 35.8 | 17.7 | 181.1 | 1974-2015 | 42 |
| Rhine–Village Neuf/Weil | WM | 36'472 | 1'100 | 1'353 | 914 | 3.6 | 30 | 43 | 11 | 31.5 | 20.6 | 207.5 | 1977-2015 | 39 |
| Rhine – Rekingen | RE | 14'718 | 1'260 | 1'262 | 947 | 3.9 | 27 | 60 | - | 30.1 | 24.9 | 188.1 | 1975-2015 | 41 |
| Aare – Hagneck | HA | 5'104 | 1'370 | 1'506 | 1'106 | 2.1 | 25 | 52 | 23 | 23.9 | 29.2 | 147.3 | 1977-1982 1988-1990 1994-1996 2003-2015 | 13 |
| Rhone – Chancy | CH | 10'323 | 1'580 | 1'335 | 1'042 | 5.8 | - | 77 | 10 | 14.4 | 23.9 | 167.9 | 1977-1982 1986-2015 | 30 |
| Lümpenenbach | LU | 0.94 | 1'300 | 2'127 | 1'879 | 0 | - | 100 | - | 21.3 | 55.8 | 0 | 2005-2015 | 11 |
| Rhine - Diepoldsau | DI | 6'119 | 1'800 | 1'319 | 1'196 | 0.4 | - | 100 | - | 8 | 46.9 | 54.9 | 1976-2015 | 40 |
| Rhone - Porte du Scex | PO | 5'244 | 2'130 | 1'372 | 1'101 | 0.4 | - | 100 | - | 6.1 | 31.7 | 58.5 | 1974-2015 | 42 |
| Inn - S Chanf | SA | 618 | 2'466 | 1'063 | 1'036 | 1.6 | - | 100 | - | 3.3 | 43 | 27.5 | 1998-2015 | 18 |
| Erlenbach | ER | 0.76 | 1'300 | 2'182 | 1'660 | 0 | - | 100 | - | 2.9 | 52.5 | 0 | 2005-2015 | 11 |

*Table 2:* Results of the C-Q relations analysis. The symbols "+","−" and "=" refer to the possible behavior combinations described in Figure 7, while the numbers indicate how many catchments exhibit a specific behavior for each solute. The solutes are classified as reported in the first column.

| Solute class | Solute | Behavior | | | | | | | | |
|---|---|---|---|---|---|---|---|---|---|---|
| | | +/+ | +/= | +/- | =/+ | =/= | =/- | -/+ | -/= | -/- |
| Geogenic solutes | $Ca^{2+}$ | 0 | 0 | 0 | 0 | 1 | 1 | 0 | 1 | 8 |
| | $Mg^{2+}$ | 0 | 0 | 0 | 0 | 0 | 0 | 0 | 0 | 11 |
| | $Na^{2+}$ | 0 | 0 | 0 | 0 | 0 | 0 | 0 | 0 | 11 |
| | $H_4SiO_4$ | 1 | 1 | 0 | 1 | 1 | 2 | 0 | 0 | 5 |
| | $K^{2+}$ | 0 | 0 | 0 | 0 | 0 | 0 | 0 | 0 | 11 |
| Deposition derived | $Cl^-$ | 0 | 0 | 0 | 0 | 0 | 0 | 0 | 1 | 10 |
| Nitrogen species | $NO_3$ | 0 | 0 | 0 | 0 | 2 | 0 | 0 | 2 | 7 |
| | TN | 0 | 1 | 0 | 0 | 2 | 0 | 0 | 5 | 3 |
| Phosphorus species | DRP | 0 | 0 | 0 | 1 | 2 | 1 | 0 | 3 | 4 |
| | TP | 2 | 1 | 0 | 5 | 0 | 0 | 3 | 0 | 0 |
| Organic Carbon species | DOC | 0 | 3 | 0 | 1 | 5 | 0 | 0 | 0 | 2 |
| | TOC | 6 | 1 | 0 | 4 | 0 | 0 | 0 | 0 | 0 |
| Total (%) | | 6.8 | 5.3 | 0 | 9.1 | 9.8 | 3.0 | 2.3 | 9.1 | 54.5 |

**Figures**

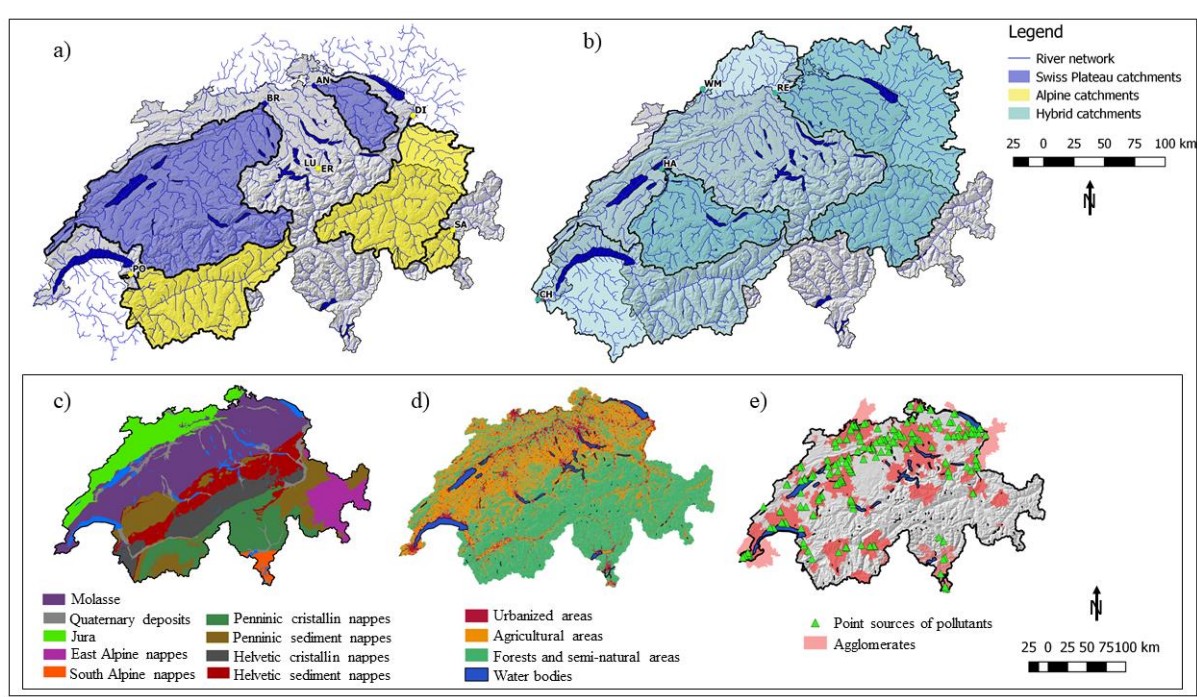

*Figure 1:* Map of NADUF monitoring stations and description of the study sites. The upper panel represents the study sites. a) Swiss Plateau (blue) and the Alpine catchments (yellow), b) the catchments spanning both regions, hybrid catchments (light blue). The bottom panel describes the study sites in terms of c) macro-geological classes, d) land cover and e) anthropic pressure.

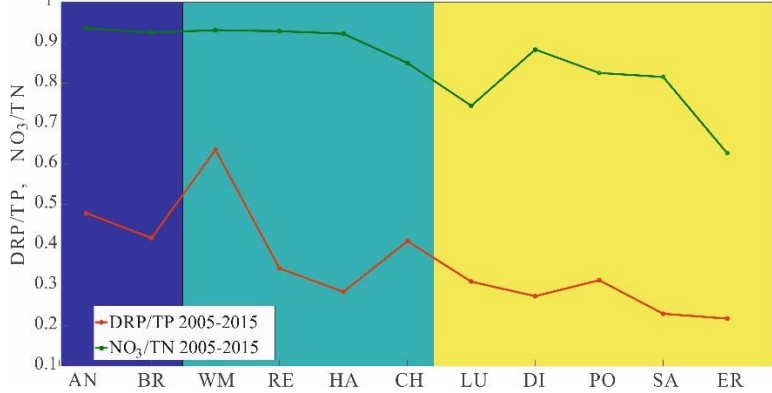

***Figure 2.*** Ratios of DRP/TP (red) and NO₃/TN (green) across catchments computed on the period 2005-2015.
Both the patterns show a decreasing trend from more to less anthropogenically affected catchments (left-to-right
of x axes). This pattern is more evident for phosphorus. Background colors refer to the catchment classification
explained in Session 3.1.

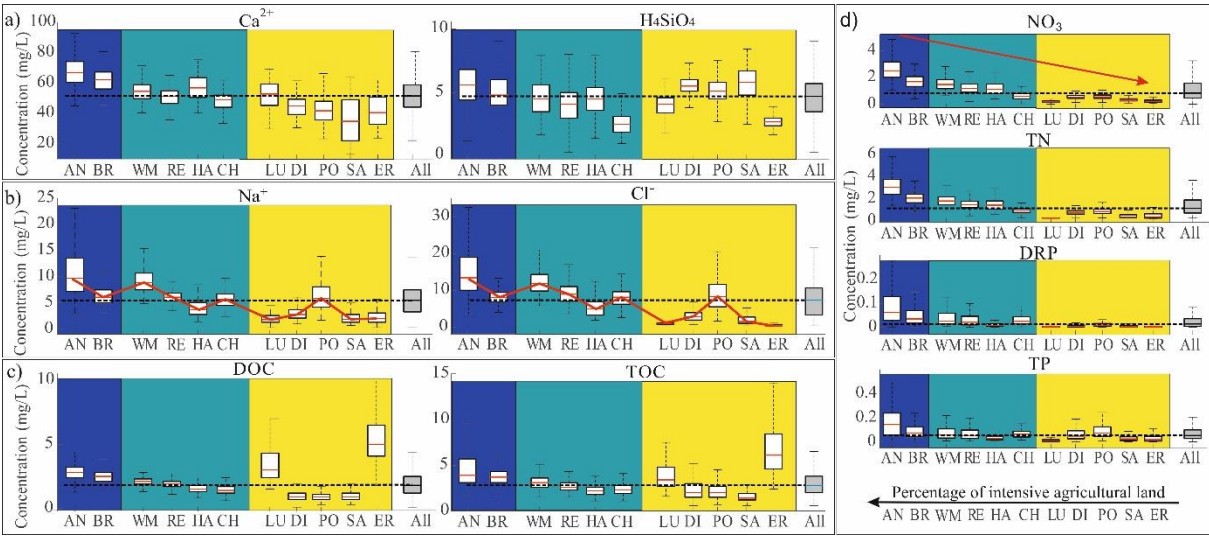


***Figure 3:*** Boxplot of measured concentrations across catchments. The grey box on the right of each subplot refers
to the concentrations computed from all the observations of all the catchments. The black horizontal dashed line
represents the median of all the measurements across all the catchments. Panel a) shows the effect of bedrock
geological composition on Ca²⁺ and H₄SiO₄ concentrations. Panel b) shows the pattern of Na⁺ and Cl⁻
concentrations across catchments. Panel c) shows the DOC and TOC concentrations. Panel d) shows the decreasing
trend of nutrients median concentrations. The catchments are ordered by increasing percentage of land used for
intensive agriculture, as shown in the bottom table and the background colors refer to the catchment classes: Swiss
Plateau (blue), hybrid (light blue) and Alpine (yellow) catchments.

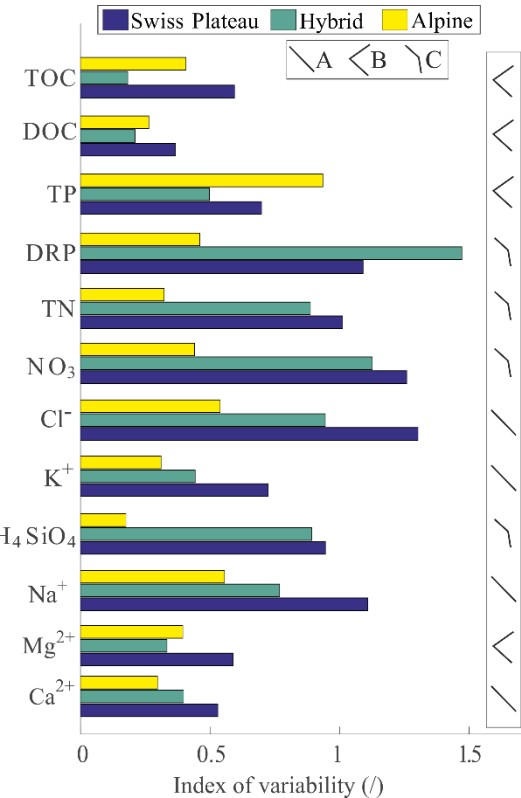


*Figure 4:* Bar plot of the index of variability. Each bar represents the average monthly variability of average concentration relatively to discharge variability per catchment class. The colors of the bars differentiate catchment morphologies: blue for Swiss Plateau, aqua-green for hybrid and yellow for Alpine catchments. The A, B and C represent the observable patterns of the index of variability across the three classes. Type A is the result of the different seasonality of discharge dominating the response. Type B refers to those solutes with an index of variability much higher in the Alpine catchments than in the others. Type C represents solutes with the index of variability higher in Swiss Plateau catchments than in the other classes.

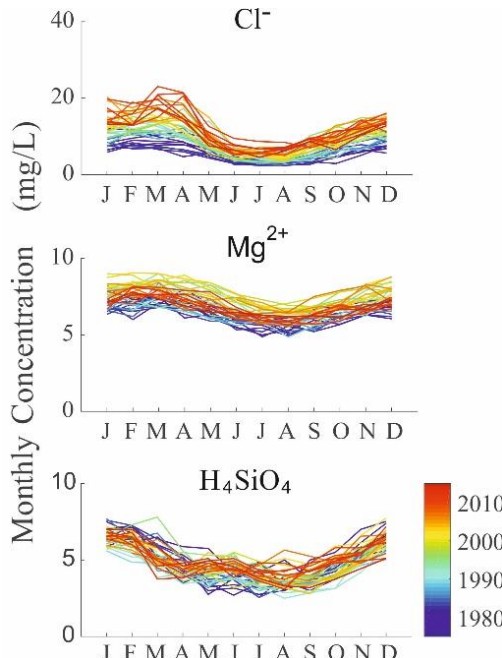

*Figure 5:* Three exemplary long-term patterns of solute concentrations. The upper box represent a clear
increasing trend, the middle box a non-monotonic trend (firstly increasing and then decreasing), while the
bottom box shows the absence of any trend. The patterns are shown for the station of Aare − Brugg as an
exemplary case.

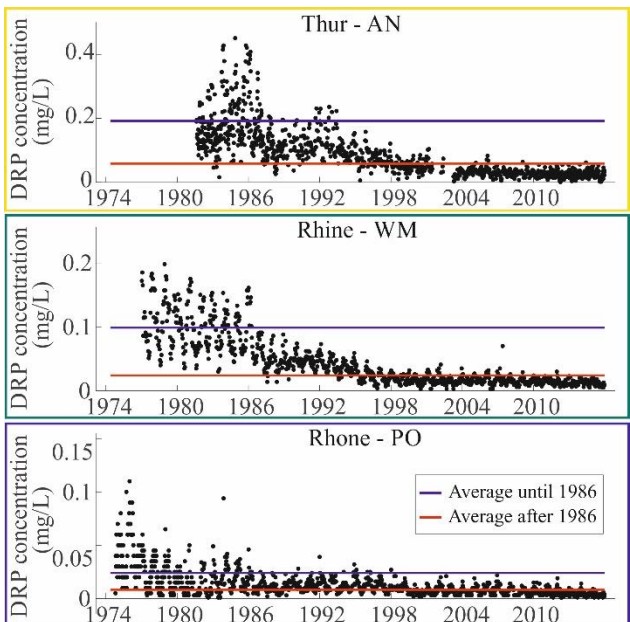


**Figure 6.** Observed DRP concentrations in three catchments characterized by different classes (i.e. Thur-AN, Rhine-WM, Rhone-PO). The blue line represents the mean until 1986, whereas the red line represents the mean after 1986 and until the end of the monitoring period. After the introduction of the phosphate ban in 1986, the DRP concentrations have shown an evident decrease.

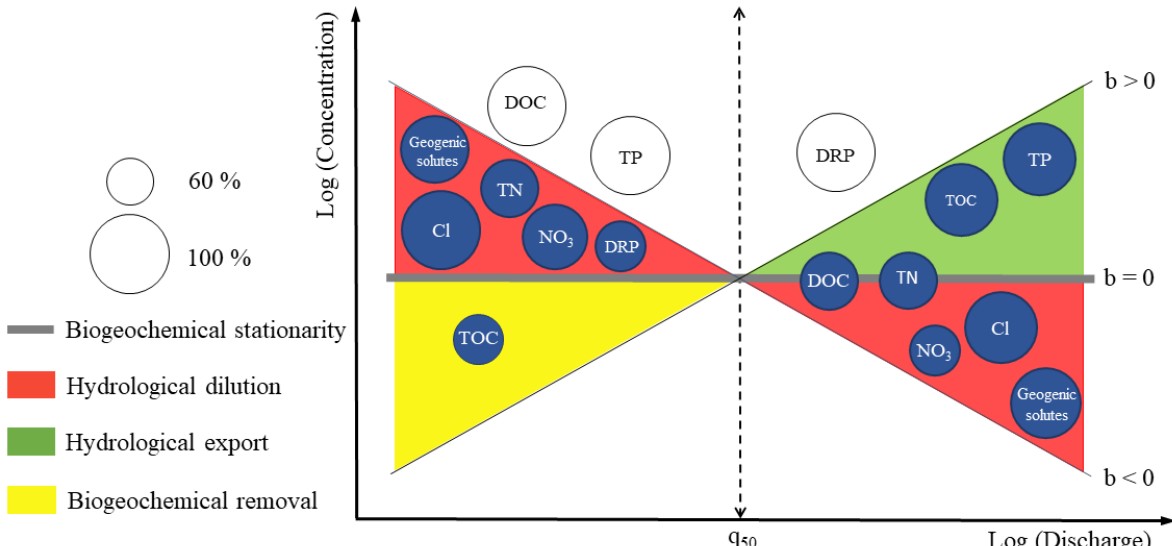

954

**Figure 7:** Solute behaviors classification in the log(C)-log(Q) space. The definitions are derived from the classification of *Moatar et al.* (2017), which is based on the value of *b*, the slope of the regression line in the log(C)-log(Q) space. Discharge time series is divided in low-flow and high-flow events based on $q_{50}$ the median daily discharge. Red areas represent hydrological dilution behavior, yellow areas represent biogeochemical removal for low flows, while green areas represent hydrological export behavior. The grey horizontal line crossing the axes origin represents the near-zero slope area, i.e., it is representative of biogeochemical stationarity. The colorless solutes outside these areas do not show any dominant behavior. The dimension of circles represents the percentage of catchments in which the dominant behavior is observed (from 60 to 100%).

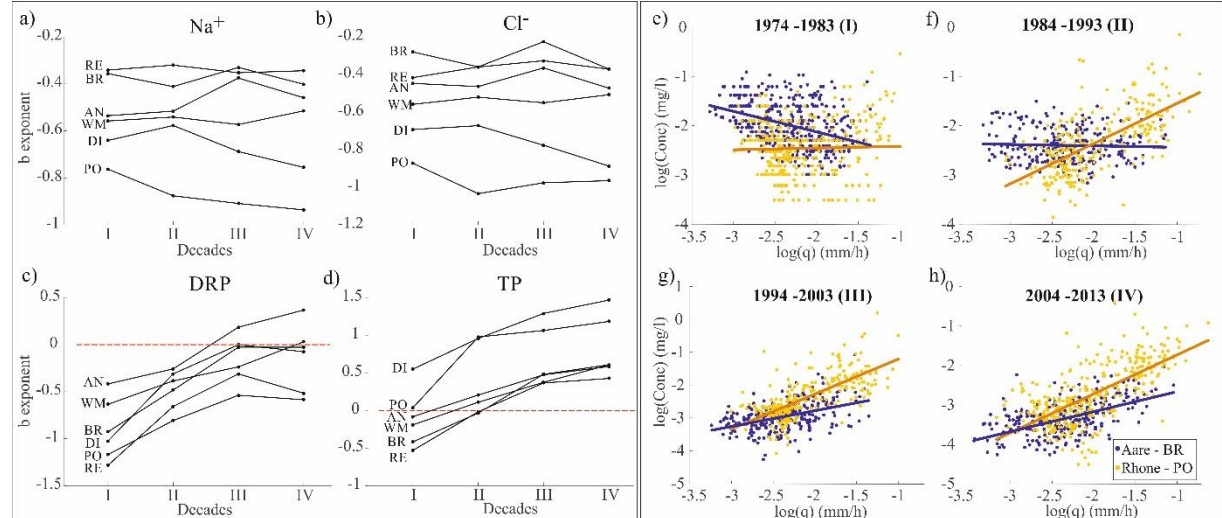


*Figure 8:* Analysis of temporal variations of the b exponent. The left plots represent the values of b exponent of
the C-Q empirical relation (C =aQ$^b$) of a) Na$^+$, b) Cl$^-$, c) DRP and d) TP across four decades from 1974 to 2013
((i) 1974-1983, (ii) 1984-1993, (iii) 1994-2003 and (iv) 2004-2013) across all the catchments with monitoring
period longer than 30 years. The dashed red line represents the zero threshold (i.e., biogeochemical stationarity).
The right panel represents two examples of how the C-Q relations vary across the decades e) 1974-1983, f) 1984-
1993, g) 1994-2003 and h) 2004-2013. The C-Q relations refer to the catchments BR (Swiss Plateau, in blue) and
PO (Alpine, in yellow) for the total phosphorus.