# Peer review of "Anthropogenic and catchment characteristic signatures in the water quality of Swiss rivers: a quantitative assessment"

_Hydrology and Earth System Sciences, 2018_

## Referee Comment (RC1) · Anonymous Referee #1 · 7 Jun 2018

**General impression**

Botter et al. examines in the manuscript 'Anthropogenic and catchment characteristic signatures in the water quality of Swiss rivers: a quantitative assessment' the dataset of the Swiss National River and Survey Program (NADUF). The dataset consisting of biweekly water samples collected in different catchments throughout Switzerland, which were analysed on wide range of different chemical variables. The authors represented the different variables of eleven catchments as boxplots, regime type, variability index, temporal representation and concentration-discharge relations to infer and generalize the impact of catchment characteristics and human activities. The separation of the data into low and high flows gives a new view on the data.

The manuscript is structured but contains some inconsistencies, is sometime not clear or confusing. Definitions need to be better explained, e.g. anthropogenic, human influence and intensive and extensive agriculture in the introduction. Findings need to be reported consistently, e.g. section 4.1 phosphorous or silica L250 is coming from fertilizers/humans while section 5.1 and L283 reports that it might be also due erosion or geology. Related to this, chose the appropriate data and analysis to answer the research question.

Not coming from Switzerland, it is hard to understand where the metropolitan areas are, which kind of land use and land cover or geology the different catchments have and how this affects the water quality. Therefore adding such information in Figure 1 will help to interpret the data, e.g. why certain catchment have higher Ca, NO3 or DOC concentrations compared to others.

In addition to the defined objectives, it might help to phrase a clear research question and state which the different hypothesis are. This will allow to distinguish between different processes and the complex interaction of climate, catchment characteristics (land use and land cover, geology, topography, shape…) and humans affecting stream chemistry. The influence of climate forcing on chemical variables was only vaguely discussed and not supported through any analysis, but it is required in the next version.

The number of figures in the main article and supplementary material is overwhelming. By refereeing to both main article and supplementary material, results in that the reader get lost and forgets about the main findings. Figures contain lots of information, but the results are not equally explained for every figure. With regard to the figures, I suggest moving important figures into the main article and leaving "less" important ones in the supplementary material. To decrease the number of figures and focus on the main findings different figures could be combined, e.g. Figure 2 and 10, leave out Figure 3 (almost similar information as Figure 4?) and combine Figure 5 and 7.

In the presented work, a subset with consistent temporal data was used. However, it would be unfortunate to exclude how the stream chemistry changes in space. The full data set could be used to perform an additional spatial analysis to infer at which scale the effects of climatological forcing, catchment characteristics and human influence can be detected.

It is necessary to correlate the chemistry with other variables than agricultural land use as land cover, geology, urban area etc. To strengthen the findings it is necessary to perform a comparison between variables and catchments and test whether the findings of Figure 4, 6, 7 and 9 are significant different. In addition, to be able to judge the validity of the results and limitations of the dataset, it is necessary to consider measurement errors and detection limits of chemical variables.

The discussion and conclusion sections are wordy, without highlighting the new findings. To explain the data, different streamflow generation and runoff processes were hypothesized to occur. However, the discussion section lacks debating the temporal sampling interval and its ability describe potential occurring processes at shorter time scales. In addition, it is necessary to refer to the current understanding of streamflow generation, runoff processes and stream network connectivity observed around the world. But also studies from Switzerland, which is known as a country to have a long, history and good knowledge of detailed small-scale catchment processes understanding (from lowlands to Alpine regions), needs to be included. The authors need to revise this section and including these points to be able to put findings into a spatial and temporal perspective.

The manuscript will benefit by streamlining the discussion and conclusion with focus on the effects of climatological forcing, catchment characteristics and human influence on the spatial and temporal variability of chemical variables. It would be also valuable to add a discussion section, from a hydrologist – scientific point of view with an outlook for potential next steps or a critical note on the data collection and what could / should be done differently to address certain future questions.

**Specific comments referring to line**

| 27 | "*Both natural and anthropogenic factors impact...* " If this is the case, it is necessary to perform an appropriate analysis to separate the different factors. |
|---|---|

36, 52    "*… residence time...*" residence times are usually calculated using conservative tracer which is not part of this study. Please clarify.

51    If the catchment structure is important did you test for this?

94    Please clarify the focus of the study, is it human or anthropogenic. How do you distinguish between human and non-human influences?

100    In the table, please indicate the start and end of the available data of the different catchments. What is the impact of the length of different time series? It could be that different decades where drier compared to others. This could affect the results. Please test and explain.

102-103    Please rephrase "basins" and "*large catchments*". Especially since ER LU are rather small catchments.

109    Here you could highlight the long-term character of the data.

116-119    This section is general. Please be more specific and refer to figures.

117    Please explain what "anthropogenic pressure" is.

120    In the discussion section please discuss how the temporal sampling could affect your results. If these samples are long stored, could reactive processes alter the sample and affect the results? Please comment on this.

130,222,250    Phosphor but also other variables could be also due to weathering. Please comment on this.

134    Why are these variables available and not shown or used in the analysis?

147    Please explain why this classification was used and not a classification based on the regime type, precipitation, geology, or land use.

148    Is there a different term for hybrid catchments?  It sounds like a mythological creature.

149-152    This sentence is a conclusion. Please move in the appropriate section.

203    Please explain these ratios.

219    Whys is this not surprising? Please explain better or rephrase the sentence.

230    "*...seasonality of streamflow Swiss Plateau catchments is determined by a combination of precipitation and snowmelt.*" Isn't this the case in pre and or Alpine catchments? Please explain.

249    " *… there are factors...*" which factors please specify.

307    "*Solute export across catchments seems to be mostly controlled by anthropogenic factors rather than by catchment characteristics.*" None of the analysis supports this conclusion! Please perform an analysis, which separates the effect of climatology, geology and human.

313-314    How representative is such a ratio for agriculture seen ratio of the plateau catchments and e.g. LU is similar? Please comment on this.

316    This statement needs to be better explained. What are the differences and which other analysis were used?

319-320    Please provide a back of the envelope calculation to support this statement.

324 -327    Consider moving Figure S1 into the main document to show the signal.

Please explain the definition low intensive (one cow) vs. high intensive (several  cows?). In winter cattle is kept indoors while in summer outdoors. Can you observe this in your data?

If processes overlap, how is it possible to distinguish from dominant processes and natural vs. human processes? Please comment on this.

| | |
|---|---|
| L330-331 | Please clearly state the difference between agriculture and human activities. Also, in terms of the type of signal one could expect. |
| 333-338 | This section is not clear. Please guide the reader through Figure S2. In which way does the coefficient de Pardé of discharge and loads relate to each other? Please explain. |
| 354 | Throughout the document it is necessary to clearly state what are areal sources e.g. cities, point sources e.g. Water treatment plants, line sources e.g. streets and what the implications are on the chemical signal. This because data at the catchment outlet might not be representative for all of the catchment. Please discuss this critically. |
| 363 | If this figure is important and discussed, it should be in the main manuscript. It would be also interesting to add a "natural" catchment to show the difference between pristine and human influence catchment. |
| 367-370 | This sentence is not clear, what are secondary effects. Could a Ca/Mg ratio be useful to show whether the effect is really from fertilizers and not from weathering? It also is necessary to provide data on what would be a natural background value and what could additionally come from fertilizers. |
| 378 | This seems a hypothesis and not supported by your analysis. |
| 379 386 | By including a geology map and other catchment characteristics will reveal which catchments are affected and by which percentage! Please add and comment. |
| 387 | This sentence is not clear, phrasing and hydrological processes. Atmospheric forcing causes also a temporal variability in steep and flat catchments. Please comment / explain. |
| 389 | Also in bare fields, soil erosion can be high. Can you see this in the data? |
| 396-414 | This section is not clear. In the "alpine" categorized catchments, not every catchment has a glacier. Please comment and explain. This is where global and local runoff processes understanding need to be added. |
| 420-426 | I believe that lakes dampen the signal, but where is the analysis to support this statement? In addition, how do other processes, e.g. instream processes, change the signal of interest? |
| 396-414,438-443 | Including recent findings on streamflow generation and runoff processes (e.g. America, Europe and Japan) might help to explain the different signals and improves the discussion. |
| 476 | Please define what low-flow is. Does this differ among catchments and how does it affect concentrations. Please comment. |
| 496 | Why are the atmospheric forcing and catchment characteristics less evident? Based on what? Please explain and rephrase. |
| 499 | Despite this spatially rich dataset, how many samples e.g. catchments would one need. What is the effect of scaling in your signal and how would this effect your results - macro pattern vs. micro pattern? |
| Table 1 | In the caption there is written north-south gradient. Is this gradient geographically or rank in the table? Please clarify. Maybe change in the header row the names to - mean annual precipitation and mean annual discharge? |
| Figure 1 | To better link the different catchments use a consistent color scheme as in figure 6 |
| | Please add basic information such as scale bar, north and legend. |
| | Also, add information on land cover and land use and large cities, geology and country names. |
| | It is difficult to see where lakes are and why streams, crossing the Swiss borders , are not represented. |
| Figure 2 | To help the reader please specify in caption what $b$ and q50 means. |
| Figure 3 | Why do some variables have species in their name and other not? Maybe add also the CV to compare different chemical variables. Due to similarities with figure 4, this figure could be removed. |
| Figure 4 | Give boxplots same color of catchment classification used in figure 6. |
| | Which catchment is **WM**? In Figure 1, this catchment is not visible and might be a typo? |

In d) one could argue on an existing decreasing trend with decreasing percentage of agriculture. This due to e.g. outliers e.g. BR-HA where after the information seems to flatten out. Please provide statistical test and in addition, a significant test to compare the different catchments and support your statement.

In addition, is there a real difference in the light of measurement accuracy?

Generally, are these concentrations and their variability high or low? Maybe compare the concentrations with observations elsewhere and report in the discussion.

Figure 5    Why the different streams were clustered according to a hydrological regimes classification? Do the different regimes really cluster e.g. only Inn and Rhine seem to be similar regimes but others not. Please explain.

Caption: "*Each point represents the monthly average discharge ...*" should not this be a ratio as described in the y-axis?

"*Hybrid catchment*" Maybe choose a different term.

Figure 6    Please be consistent with the terms. Is it index of variability or variability index?

"*...discharge variability per catchment.*" This should be per catchment class.

Rank the catchments as classes similar to Figure 5.

Next to referring to section 3.2, please explain shortly the patterns and add a legend.

Figure 7, 262    Please better explain in the result section the signals and processes.

The y-axis should be labeled as mean monthly concentration.

Why was only the station at Rekingen shown? It would be interesting to see other catchments Alpine vs Urban influenced catchments and compare the change in amplitudes or signals.

Figure 9    Please add labels to each plot a-d and specify which line is which catchment.

Right panels) Why was only the station at Rekingen shown? It would be interesting to see other catchments Alpine vs Urban influenced catchments and compare signals.

Figure S1    The labels and captions are not coherent and make it difficult to understand. Please change and double check all figures.

Figure S2    …" *Monthly average of discharge...*"  The caption and y-label are not coherent and confuse the reader.

---

## Referee Comment (RC2) · Anonymous Referee #2 · 20 Jul 2018

Anthropogenic and catchment characteristic signatures in the water quality of Swiss rivers: a quantitative assessment Martina Botter, Paolo Burlando, Simone Fatichi Several recent papers studied long-term series of water quality and discharge aiming to generalize behaviors of selected solutes across catchments in order to infer anthropogenic and catchment characteristic influences. This study provides some more results on Alpine streams. The authors analyzed geogenic solutes, chloride, nitrogen, phosphorus and organic carbon species, monitored by the Swiss National River and Survey Program for 11 Swiss rivers with a temporal resolution of 14 days as composite sampling (sampling represent an integration of the preceding 14 days) for more than 10 years. The analysis of basic statistics, seasonality, temporal trends and concentrationdischarge behavior revealed impacts of human activities for some catchments. However, the influence of catchment characteristics is much less evident. This is probably due to the small number of analyzed catchments and to their area range which is very bi-modal (one group with catchment area around 5000 to 30 000 $km^2$ vs. 2 small catchments with area < 1$km^2$) which do not help having a more quantitative spatial analysis. The manuscript needs to better explain the relation between temporal metrics and spatial characteristics. Another way of analyzing the results could be to consider the variation of these relationships along nested catchments (Rhein, Rhone, Aare). The manuscript has a relatively good structure, but the results could be presented in a more factual way, in order to better distinguish them from the discussion. The conclusion needs to highlight the new findings of this work. Database and study sites The authors do not present very well the database (numbers of data/years for each site and element, screening, discussion about the difference between composite sampling and grab sampling, representativeness of metrics calculated from composite sampling, especially for small catchments). It is not clear either whether all the calculated temporal metrics are based on mean bi-monthly concentration and discharge data time series. If this is the case, the authors need to discuss how this sampling design impacts the analysis of the temporal metrics (especially concentration-discharge relationships). Catchment characteristics are not very well presented. Figure 1 could be reworked to present land use/land cover. Colors for catchment could be replaced by contour lines. For example, authors defined three categories of catchments according to their morphology and geographical locations (lines 148) but it is not clear why only these criteria. It seems that these regions are homogenous also for land use, lithology and climate? Hence, do they belong to the same hydro-ecoregion? It might help to see on figure 1 or in table 1 theses three categories (how many catchments for each category) to link them to geology, landuse/land cover. Table 1. Please use $km^2$ as unit for catchment size, and specific discharge (l s-1 km-2) for discharge, also in figures (ex. Figure S5,) in order to allow catchment comparisons. ID=VW on table 1 but ID=WM on figure 1. Is it the same catchment? Temporal metrics: it is not very clear what is

the aim of each indicator, especially for the seasonality and C-Q relationship. Index of "seasonal" variability: the numerator of the equation could be reformulated to take into account that it performs a sum of deviations for different catchments belonging to the specific "topographic" class. It is consistent for all "topographic" classes with only 3 to 4 catchments in a category ? Figure 5. How hydrological regimes were defined? The method is not presented in chapter 3. What is the link with Figure 6 (index of seasonal concentration variability), and with figure S2 ? Concentration-discharge relationship. Please define why you calculate integral "b" exponent and truncated "b" exponent, b50sup, b50inf. Figure 2 and Figure 10 can be merged, indicating that you use the conceptual diagram of C-Q relationships proposed by Moatar et al, 2017 and test it for Swiss rivers (mean altitudes > 1000 m, mean rainfall 1000 - 2000 mm/y). You can also compare with other recent papers (ex. Diamond, Cohen, 2017 for coastal Plain Rivers in Florida) In the split-hydrograph method, separate concentration-discharge relationships are described for below and above median discharge, Q50 is the median of daily discharge. Are your C-Q diagrams (Figure 9, 10) realized from mean bi-monthly concentration with mean bi-monthly discharge? It would be the reason why only 29% of the catchment-solute combinations have different behaviors between low- and high-flow conditions. Or perhaps, it is a characteristic of alpine rivers where dilutions and exports of elements are the major behaviors. While biogeochemical and retention removal processes at low flows are not very significant. Or perhaps, Q50 is not the appropriate discharge percentile break-point? Figure 8. What site? Figure 6. A, B, C not defined in section 3.2 Figure 10. define grey areas

---

## Author Comment (AC1) · 3 Aug 2018

**Reply to Referee 1**

**Botter et al. examines in the manuscript 'Anthropogenic and catchment characteristic signatures in the water quality of Swiss rivers: a quantitative assessment' the dataset of the Swiss National River and Survey Program (NADUF). The dataset consisting of biweekly water samples collected in different catchments throughout Switzerland, which were analysed on wide range of different chemical variables. The authors represented the different variables of eleven catchments as boxplots, regime type, variability index, temporal representation and concentration-discharge relations to infer and generalize the impact of catchment characteristics and human activities. The separation of the data into low and high flows gives a new view on the data.**

**The manuscript is structured but contains some inconsistencies, is sometime not clear or confusing.**

We thank the reviewer for the interest in the paper and of the constructive criticism of our study. We will work on the first version for enhancing the clarity and consistency.

**Definitions need to be better explained, e.g. anthropogenic, human influence and intensive and extensive agriculture in the introduction.**

The definition of these terms is fundamental for a straightforward understanding of the analysis and of the results. We thank the reviewer for pointing out the inaccuracy in the explanations. We will give clearer definitions in the next version of the manuscript.

**Findings need to be reported consistently, e.g. section 4.1 phosphorous or silica L250 is coming from fertilizers/humans while section 5.1 and L283 reports that it might be also due erosion or geology. Related to this, chose the appropriate data and analysis to answer the research question.**

We thank the reviewer for pointing out this inconsistency, which occurred due to an improper or inconsistent use of the term "input" across the manuscript. The sources of nutrients (e.g., phosphorus, nitrate,..) intended as input into the catchment system are often fertilisers. Once in the catchment, these solutes are transported and can be transformed in the different compartments of the catchments. Considering phosphorus, as an example, consolidated knowledge in literature states that it tends to accumulate in the soil where it can sorb, desorb, be mineralised or immobilised, and only a minimal fraction leaches into the groundwater. When soil is eroded, phosphorus-rich soil particles are taken by flowing water and therefore soil erosion is one of the main contributor to the phosphorus load into the rivers. The two processes are strictly connected and the contribution of one does not exclude the other. This point will be better explained in the manuscript.

**Not coming from Switzerland, it is hard to understand where the metropolitan areas are, which kind of land use and land cover or geology the different catchments have and how this affects the water quality. Therefore adding such information in Figure 1 will help to interpret the data, e.g. why certain catchment have higher Ca, NO3 or DOC concentrations compared to others.**

We agree with the reviewer that the case study description was lacking of some fundamental information and that Table 1 and Figure 1 do not explain exhaustively the catchments. We will add a more extensive description including land use, land cover, and geology information in section 2 and we will find a more suitable graphical representation of Figure 1.

**In addition to the defined objectives, it might help to phrase a clear research question and state which the different hypothesis are. This will allow to distinguish between different processes and the complex interaction of climate, catchment characteristics (land use and land cover, geology, topography, shape…) and humans affecting stream chemistry.**

We will re-phrase the research questions in the introduction to clearly state what we are investigating, concerning hypothesis, being this a "data-driven" analysis and not a numerical or field experiment we prefer to refrain from formulating a posteriori hypotheses, given also the challenge to test them with 11 catchments only.

**The influence of climate forcing on chemical variables was only vaguely discussed and not supported through any analysis, but it is required in the next version.**

We thank the Reviewer for this observation. In the manuscript there is not a specific analysis dedicated to the influence of climate on stream biogeochemistry, because the sample does not suit the requirements of size and

independence necessary for a formal statistical analysis of climatic effects. The study is indeed based on only 11 catchments across Switzerland, often nested catchments. Table 1 gives an overview of the catchments including mean annual rainfall, which could represent the major source of climatic variability across the catchments being other variables as temperature, humidity and solar radiation rather similar at catchment scale. However, the mean annual rainfall across the catchments vary in a limited range 1063 – 1506 mm/y; the only exceptions are Lümpenenbach (LU, 2127 mm/y) and Erlenbach (ER, 2182 mm/y) catchments, situated in the Alptal valley. These two catchments are also the two smallest catchments considered in the study, with an area at least three orders of magnitude  smaller than the others and therefore some observed behaviours might be due to the wetter conditions but also due to the smaller size. The effects of the two factors cannot discerned with such a small sample size and any dedicated analysis to partition climatic effects is not meaningful given the similarity in climate of all the other basins.

**The number of figures in the main article and supplementary material is overwhelming. By refereeing to both main article and supplementary material, results in that the reader get lost and forgets about the main findings.**

The number of figures will be reduced and some figures will be moved from the main article to the supplementary material and vice versa. Some of these modifications include moving Figure S4 from supplementary information to the main manuscript, removing Figure 3 and merging Figure 2 with Figure 10.

**Figures contain lots of information, but the results are not equally explained for every figure.**

We will be more precise in referring results to the figures and we will reduce the information content of each figure so that it matches only the one required for the discussion. For example, Figure 3 is not crucial for the discussion and Figure 7 contains much more information than the one discussed in the article. The reviewer's suggestion of combining figures 5 and 7 will be taken into account.

**With regard to the figures, I suggest moving important figures into the main article and leaving "less" important ones in the supplementary material.**

As explained above, we will take into consideration the suggestion of moving Figure S4 to the main article and leave Figure 3 either out of the article or in the supplementary material.

**To decrease the number of figures and focus on the main findings different figures could be combined, e.g. Figure 2 and 10, leave out Figure 3 (almost similar information as Figure 4?) and combine Figure 5 and 7.**

We thank the Reviewer for these suggestions. Please see our replies above.

**In the presented work, a subset with consistent temporal data was used. However, it would be unfortunate to exclude how the stream chemistry changes in space. The full data set could be used to perform an additional spatial analysis to infer at which scale the effects of climatological forcing, catchment characteristics and human influence can be detected.**

The objective of our study is to analyse long-term water quality data in order to investigate the possible signature of anthropic activity and of catchment characteristics. The focus of this article is on purpose on the temporal analysis more than on the spatial variability and on time-series of a length that is deemed sufficient to filter short-term variability and average long-term behaviours. An exhaustive spatial analysis would require different criteria for station selection, different type of analysis, in other words a very different study with another database, which we consider to be out of the scope of our study. The eleven selected catchments, meaning water quality samples in eleven points across the entire Switzerland provide some information about spatial variability that is included in the narrative of the article, but are not sufficient for an extensive and statistically robust analysis of spatial variability.

**It is necessary to correlate the chemistry with other variables than agricultural land use as land cover, geology, urban area etc. To strengthen the findings it is necessary to perform a comparison between variables and catchments and test whether the findings of Figure 4, 6, 7 and 9 are significant different.**

The study will be modified to benefit of the integration of other variables like land cover, geology and urban areas. We thank the reviewer for the suggestion of testing the correlation between these variables and the catchment behaviours. Other studies (e.g., *Godsey et al.*, 2009; *Moatar at al.*, 2017) apply the Spearman's non parametric test (*Spearman*, 1904) to this purpose. This test quantifies the correlation between the b exponent, representing the

biogeochemical response of the catchment, and the variables, representing the different catchment characteristics. We are planning to integrate this analysis in our work, thus supporting the discussion with more quantitative results. However, please note that we are considering 11 catchments only.

**In addition, to be able to judge the validity of the results and limitations of the dataset, it is necessary to consider measurement errors and detection limits of chemical variables.**

We agree with the reviewer that the manuscript is lacking comments concerning measurement errors and detection limits of chemical values. Concerning the detection limits, we considered them during the preliminary data elaborations, when the database was pre-processed to clean outliers due to issues in detection limits. This was not mentioned in the manuscript and it must be integrated in the revised version.

**The discussion and conclusion sections are wordy, without highlighting the new findings.**

In the revised manuscript, the discussion and conclusion sections will be more essential and the main findings will be pointed out clearly.

**To explain the data, different streamflow generation and runoff processes were hypothesized to occur. However, the discussion section lacks debating the temporal sampling interval and its ability describe potential occurring processes at shorter time scales.**

This is an important observation and we agree that the discussion should include statements on the ability of the temporal sampling interval to describe potential occurring processes at the shorter time scale. The sampling method of the NADUF database is flow-proportional and it is characterised by a 14-days frequency. The low sampling frequency does not allow recognising short-time scale signals in the biogeochemical signature of rivers, even though these are integrated in the 14-days frequency, being the sampling "flow-proportional". Water quality programmes established in many catchments around the world record data at quarterly, monthly, 14-days or weekly frequency because of time and financial constraints. Despite this low sampling frequency, we think that these data have a high information content if used in the proper way. The representation of short time scales processes cannot be the goal of this kind of database, but the description of long-term patterns of water quality parameters in relation to external forcings (i.e., anthropic activities and catchment characteristics) can. Studies addressing the comparison between information at different time resolutions are cited in the manuscript as alternative to our approach (*Butturini et al.*, 2008; *Duncan et al.*, 2017; *Minaudo et al.*, 2017; *Wade et al.*, 2012).

**In addition, it is necessary to refer to the current understanding of streamflow generation, runoff processes and stream network connectivity observed around the world. But also studies from Switzerland, which is known as a country to have a long, history and good knowledge of detailed small-scale catchment processes understanding (from lowlands to Alpine regions), needs to be included. The authors need to revise this section and including these points to be able to put findings into a spatial and temporal perspective.**

We thank the reviewer for the suggestion and we will enrich the discussion with examples from literature focused on the hydrological processes understanding whenever it is suited.

**The manuscript will benefit by streamlining the discussion and conclusion with focus on the effects of climatological forcing, catchment characteristics and human influence on the spatial and temporal variability of chemical variables.**

Concerning climatological forcing effects and spatial variability please refer to our comments above. The discussion and conclusion sections will be modified to streamline the research objectives of the work. The discussion section is structured already in paragraphs referring to each one of the research objectives, but it can be improved making it more essential and focused on the most important results only.

**It would be also valuable to add a discussion section, from a hydrologist – scientific point of view with an outlook for potential next steps or a critical note on the data collection and what could / should be done differently to address certain future questions.**

We thank the reviewer for these suggestions. Since the data issue is widely discussed in water quality studies (e.g., *Stelzer and Likens*, 2006) we will just add a short note on data collection. A brief outlook with potential next steps will be also included in the conclusion.

Several recent papers studied long-term series of water quality and discharge aiming to generalize behaviors of selected solutes across catchments in order to infer anthropogenic and catchment characteristic influences. This study provides some more results on Alpine streams. The authors analyzed geogenic solutes, chloride, nitrogen, phosphorus and organic carbon species, monitored by the Swiss National River and Survey Program for 11 Swiss rivers with a temporal resolution of 14 days as composite sampling (sampling represent an integration of the preceding 14 days) for more than 10 years. The analysis of basic statistics, seasonality, temporal trends and concentration-discharge behavior revealed impacts of human activities for some catchments. However, the influence of catchment characteristics is much less evident. This is probably due to the small number of analyzed catchments and to their area range which is very bi-modal (one group with catchment area around 5 000 to 30 000 km2 vs. 2 small catchments with area < 1km2) which do not help having a more quantitative spatial analysis.

We agree with the reviewer that the sample of catchments is not wide enough for an exhausting representation of the spatial variability and variety of Swiss river biogeochemistry. Indeed, the main focus of our analysis is on the long-term temporal trends.

**The manuscript needs to better explain the relation between temporal metrics and spatial characteristics.**

The manuscript will be modified to include a clearer presentation of the spatial characteristics of catchments. A more exhaustive description will be integrated in Section 2. Moreover, also Figure 1 will be modified so that it can be more effective in illustrating the spatial characteristics of the catchments (e.g., urban areas, land use, geology)

**Another way of analyzing the results could be to consider the variation of these relationships along nested catchments (Rhein, Rhone, Aare).**

We thank the reviewer for the suggestion. In theory, this is a good idea, but the sample of nested catchment is limited to 3. Therefore, we are sceptical about the possibility to obtain any robust pattern or generalization originated by these three nested catchments. A detailed spatial analysis would require a different database, or better different criteria of selection of the catchments from the NADUF database favouring a higher number of stations with a much more limited number of years. Here we selected the catchments with at least 15 consecutive years of measurements for the investigation of long-term trends. For an exhaustive spatial analysis, a different selection would be necessary. However, this would require a very different methodology and it is considered to be out of the scope of our study.

**The manuscript has a relatively good structure, but the results could be presented in a more factual way, in order to better distinguish them from the discussion.**

We thank the reviewer for the positive comment and for the constructive criticism. We will present results in a more factual way. Specifically we plan to:
- Streamline the discussion section,
- Better link the results and observations in the figure,
- Eliminate the redundant or excessive information in the figures,

**The conclusion needs to highlight the new findings of this work.**

We thank the reviewer for this comment and we agree that new findings need to be clearly summarised in the conclusions because they are only mentioned throughout the manuscript.

**Database and study sites**
**The authors do not present very well the database (numbers of data/years for each site and element, screening, discussion about the difference between composite sampling and grab sampling, representativeness of metrics calculated from composite sampling, especially for small catchments).**

We agree that the database can be presented more accurately. We will integrate additional information following the suggestion of the reviewer.

**It is not clear either whether all the calculated temporal metrics are based on mean bi-monthly concentration and discharge data time series. If this is the case, the authors need to discuss how this**

**sampling design impacts the analysis of the temporal metrics (especially concentration-discharge relationships).**

Yes, all the calculated temporal metrics are based on mean bi-monthly concentration and discharge. However, please note that concentration is a "flow-averaged quantity" and not a snapshot every two weeks. The only statistic computed from hourly discharge data is the median daily discharge used in the C-Q relations and this is pointed out at lines 177-181. We will introduce a sentence on how the sampling design impacts the analysis of the temporal metrics, referring to literature (e.g., *Stelzer and Likens*, 2006).

**Catchment characteristics are not very well presented. Figure 1 could be reworked to present land use/land cover. Colors for catchment could be replaced by contour lines.**

We plan to integrate section 2 with additional and more detailed information concerning the characteristics of the catchments. Specifically we will be more precise in describing the geological zones, the land use, the land cover and the urban areas. We will replace colours of catchments in Figure 1 with contour lines and, if the final result will be easier to read we will substitute the current one.

**For example, authors defined three categories of catchments according to their morphology and geographical locations (lines 148) but it is not clear why only these criteria. It seems that these regions are homogenous also for land use, lithology and climate? Hence, do they belong to the same hydro-ecoregion? It might help to see on figure 1 or in table 1 theses three categories (how many catchments for each category) to link them to geology, landuse/land cover.**

The three categories of catchments are defined for the analysis of the seasonality and the categories are based on the catchment morphology and geographical locations. Since we analyse the seasonality of in-stream concentrations in relation to the seasonality of discharge, we retain important to differentiate hydrological regimes of the catchments, which have different seasonalities. Switzerland is characterised by basically two main geographical zones, the Swiss Plateau a lowland in the north and the mountainous Alpine area in the centre and south. The two different zones have substantially different hydrological regimes (Figure 5, upper and bottom panels). However, some of the selected catchments extend in both areas and are therefore defined as "hybrid catchments". They are characterised by a seasonality, which is intermediate between the two extremes (Figure 5, central panel) and they have to be treated separately from the other two classes. The geographical sub-division of these areas is used to distinguish different hydrological regimes. It does not imply that these regions are homogeneous in terms of land use or geology. We will present more clearly the catchments characteristics in Figure 1 and we will be provide further explanations for the analysis of seasonality, pointing out that the classification is done based on the different hydrological regimes and highlighting the main conclusions:
- the seasonality of $Ca^{2+}$, $Na^{2+}$, $K^+$ and $Cl^-$ is dictated by the seasonality of discharge,
- the seasonality of $Mg^{2+}$, TP, DOC and TOC overwhelms the seasonality of discharge due to natural controls,
- the seasonality of $H_4SiO_4$, $NO_3$, TN and DRP overwhelms the seasonality of discharge due to anthropic factors (e.g., input of fertilisers).

**Table 1. Please use km2 as unit for catchment size, and specific discharge (l s-1 km-2) for discharge, also in figures (ex. Figure S5), in order to allow catchment comparisons.**

We thank the reviewer for the suggestion and we will adjust the unit in Table 1 and plot the C-Q relation with Q as specific discharge instead of discharge, however we will use "mm" rather than "l/km2", since it is a much more intuitive metric from a hydrological perspective.

**ID=VW on table 1 but ID=WM on figure 1. Is it the same catchment? Temporal metrics: it is not very clear what is the aim of each indicator, especially for the seasonality and C-Q relationship.**

Yes, this is the same catchment but the station in the original database changed the name during the monitoring period. The error originates from this inconsistency; we will use a single name throughout the manuscript so that it is going to be consistent.

**Index of "seasonal" variability: the numerator of the equation could be reformulated to take into account that it performs a sum of deviations for different catchments belonging to the specific "topographic" class. It is consistent for all "topographic" classes with only 3 to 4 catchments in a category ?**

The equation was not properly formulated since it should simply represent the average of the index of seasonal variability over catchments belonging to the same category. It will be corrected.

**Figure 5. How hydrological regimes were defined? The method is not presented in chapter 3.**

We defined the hydrological regimes based on *Weingartner and Aschwanden*, 1992. We will mention this in the manuscript.

**What is the link with Figure 6 (index of seasonal concentration variability), and with figure S2?**

The link between the classes of hydrological regime and Figure 6 is explained above. The link with Figure S2, instead, refers to one of the result we can observe in Figure 6, i.e., the seasonality of $H_4SiO_4$, $NO_3$, TN and DRP overwhelms the seasonality of discharge because of anthropic factors (e.g., input of fertilisers). Indeed, these nutrients have their own seasonality, as Figure S2 shows. In the case of $Ca^{2+}$ (bottom panel of Figure S2), instead, the pattern of the load along the year follows quite well the pattern of the discharge, also in the most human-impacted catchments indicating that external forcings is not larger than natural variability. We will emphasize this point in the revised manuscript.

**Concentration-discharge relationship. Please define why you calculate integral "b" exponent and truncated "b" exponent, b50sup, b50inf.**

We compute the truncated b exponent (i.e., $b_{50sup}$ and $b_{50inf}$) for the classification of the solute behaviours, because, as explained in lines 182-184, this allows a finer classification of their behaviours. The integral b, instead was computed to analyse how the anthropic activity influence the solute behaviours (Figure 9) beyond influencing its magnitude. The objectives of these two parts of the study are different. While the first one aims at the understanding the processes that potentially are at the basis of the observed behavior, the second one aims at detecting possible long-term trends in solute behaviour.

**Figure 2 and Figure 10 can be merged, indicating that you use the conceptual diagram of C-Q relationships proposed by Moatar et al, 2017 and test it for Swiss rivers (mean altitudes > 1000 m, mean rainfall 1000 - 2000 mm/y).**

We thank the reviewer for the suggestion and we plan to merge Figure 2 and Figure 10.

**You can also compare with other recent papers (ex. Diamond, Cohen, 2017 for coastal Plain Rivers in Florida).**

We thank the reviewer for the suggestion and we will integrate more references in the manuscript.

**In the split-hydrograph method, separate concentration-discharge relationships are described for below and above median discharge, Q50 is the median of daily discharge. Are your C-Q diagrams (Figure 9, 10) realized from mean bi-monthly concentration with mean bi-monthly discharge? It would be the reason why only 29% of the catchment-solute combinations have different behaviors between low- and highflow conditions.**

Yes, the C-Q diagrams are computed from bi-monthly concentration with bi-monthly discharge. Concentrations are flow-proportional, while the discharge is averaged on a 14-days period. The reviewer raises a very interesting point. The answer to the question is challenging since it is not possible to have a precise evaluation of which is the main factor determining different behaviours of solutes between low- and high-flows in 29% of the cases. The low sampling frequency may play a role in this. As *Stelzer and Likens* (2006) point out, sampling frequency has different effects depending on the response that concentration has to discharge, so the uncertainty related to the sampling frequency might be different from solute to solute and it is impossible to quantify it with the data available in this study. We will add a statement about the influence of sampling frequency in the discussion section.

**Or perhaps, it is a characteristic of alpine rivers where dilutions and exports of elements are the major behaviors while biogeochemical and retention removal processes at low flows are not very significant. Or perhaps, Q50 is not the appropriate discharge percentile break-point?**

We did not investigate the effects of using other metrics than Q50 as break-point. However, we think that the characteristic of alpine rivers dampen biological retention and removal processes at low flow, which are therefore not very significant.

**Figure 8. What site? Figure 6. A, B, C not defined in section 3.2**

The site is not mentioned because this is an example of the most common patterns across all of the catchments and we intentionally did not want to refer to any specific catchments. We will add in the caption the sites, the patterns are referred to, but we do not think it is an important information for the subsequent discussion.

**Figure 10. define grey areas**

We thank the reviewer for the observation and we will complete the caption of Figure 10 with the definition of the grey areas.

**References**

Butturini, A., Alvarez, M., Bernal, S., and Vazquez, E.: Diversity and temporal sequences of forms of DOC and NO3-discharge responses in an intermittent stream: Predictable or random succession?, Journal of Geophysical Research, 113, G03016, doi:10.1029/2008JG000721, 2008.

Duncan, J. M., Welty, C., Kemper, J. T., Groffman, P. M., and Band, L. E.: Dynamics of nitrate concentration-discharge patterns in a urban watershed, Water Resources Research, doi:10.1002/2017WR020500, 2017b.

Godsey, S. E., Kirchner, J. W., and Clow, D. W.: Concentration-discharge relationships reflect chemostatic characteristics of US catchments, Hydrological Processes, 23(13), 1844-1864, 2009.

Minaudo, C., Dupas, R., Gascuel-Odoux, C., Fovet, O., Mellander, P. E., Jordan, P., Shoe, M., and Moatar, F.: Nonlinear empirical modelling to estimate phosphorus exports using continuous records of turbidity and discharge, 2017.

Moatar, F., Abbott, B. W., Minaudo, C., Curie, F., and Pinay, G.: Elemental properties, hydrology, and biology interact to shape concentration-discharge curves for carbon, nutrients, sediment, and major ions, Water Resources Research, 53, 1270–1287, doi:10.1002/2016WR019635, 2017.

Spearman, C.: The proof and measurement of association between two things, The American Journal of Psychology, 1904.

Stelzer, R. S. and Likens, G. E.: Effects of sampling frequency on estimates of dissolved silica export by streams: The role of hydrological variability and concentration-discharge relationships, 2006.

Wade, A. J., Palmer-Felgate, E. J., Halliday, S. J., Skeffington, R. A., Loewenthal, M., Jarvie, H. P., Bowes, M. J., Greenway, G. M., Haswell, S. J., Bell, I. M., Joly, E., Fallatah, A., Neal, C., Williams, R. J., Gozzard, E., and Newman, J. R.: Hydrochemical processes in lowland rivers: insights from in situ, high resolution monitoring, 2012.

Weingartner, R., and Aschwanden, H.: Abflussregimes als Grundlage zur Abschätzung von Mittelwerten des Abflusses. in: Gruppe für Hydrologie, Universität Bern: Hydrologischer Atlas der Schweiz. Berne: Landeshydrologie, Bundesamt für Wasser und Geologie, plate 5.2, 1992.

---

## Author Response (AR1)

**Reply to Reviewer 1**

**Botter et al. examines in the manuscript 'Anthropogenic and catchment characteristic signatures in the water quality of Swiss rivers: a quantitative assessment' the dataset of the Swiss National River and Survey Program (NADUF). The dataset consisting of biweekly water samples collected in different catchments throughout Switzerland, which were analysed on wide range of different chemical variables. The authors represented the different variables of eleven catchments as boxplots, regime type, variability index, temporal representation and concentration-discharge relations to infer and generalize the impact of catchment characteristics and human activities. The separation of the data into low and high flows gives a new view on the data.**

**The manuscript is structured but contains some inconsistencies, is sometime not clear or confusing.**

We thank the reviewer for the interest in the paper and of the constructive criticism of our study. We modified the first version for enhancing the clarity and consistency.

**Definitions need to be better explained, e.g. anthropogenic, human influence and intensive and extensive agriculture in the introduction.**

We thank the reviewer for pointing out the inaccuracy in the explanations. We gave clearer definitions in the new version of the manuscript at L138-142 and L155-161.

**Findings need to be reported consistently, e.g. section 4.1 phosphorous or silica L250 is coming from fertilizers/humans while section 5.1 and L283 reports that it might be also due erosion or geology. Related to this, chose the appropriate data and analysis to answer the research question.**

We thank the reviewer for pointing out this inconsistency.

Concerning silicic acid, we consider it mainly sourced by rocks weathering. In L250 we state "The type C pattern, instead, **mostly** refers to human-related solutes ($H_4SiO_4$, $NO_3$, TN and DRP). The "mostly" was intended to exclude $H_4SiO_4$, since nitrogen species and DRP are human-related, but silicic acid comes from weathering. We rephrased to be more explicit on this point:

L302-304: "The type C pattern, instead, refers to solutes related to fertilization ($NO_3$, TN and DRP) and to $H_4SiO_4$, which is a product of weathering and only minimally involved in biological processes."

Concerning phosphorus, instead, the inconsistency occurred due to an improper or inconsistent use of the term "input" across the manuscript. The sources of nutrients (e.g., phosphorus, nitrate, ...) intended as input into the catchment system are often fertilizers. Once in the catchment, these solutes are transported and transformed in the different catchment compartments. Considering phosphorus, as an example, consolidated knowledge in literature states that it tends to accumulate in the soil where it can sorb, desorb, be mineralized or immobilized, and only a minimal fraction leaches into the groundwater. When soil is eroded, phosphorus-rich soil particles are taken by flowing water and therefore soil erosion is one of the main contributor to the phosphorus load into the rivers. The two processes are strictly connected and the contribution of one does not exclude the other. We added an explanation of this:

L343-346: "The label "sediment-related solutes" comes from the fact that phosphorus and organic carbon are bonded to soil particles and, when soil is eroded, carbon- and phosphorus-rich soil particles are taken by flowing water. In such conditions, soil erosion becomes one of the main contributor to the phosphorus and organic carbon load into the rivers."

**Not coming from Switzerland, it is hard to understand where the metropolitan areas are, which kind of land use and land cover or geology the different catchments have and how this affects the water quality. Therefore adding such information in Figure 1 will help to interpret the data, e.g. why certain catchment have higher Ca, NO3 or DOC concentrations compared to others.**

We agree with the reviewer that the case study description was lacking of some fundamental information and that Table 1 and Figure 1 did not explain exhaustively the catchments. We added a more extensive description including land use, land cover, and geology information in section 2 and we found a more suitable graphical representation of Figure 1. (L137-161):

[Figure]

*Figure 1:* Map of NADUF monitoring stations and description of the study sites. The upper panel represents the study sites. On the left the Swiss Plateau (blue) and the Alpine catchments (yellow), on the right the hybrid catchments (light blue). The bottom panel describes the study sites in terms of topographic areas (left), land cover (center) and anthropic pressure (right).

**In addition to the defined objectives, it might help to phrase a clear research question and state which the different hypothesis are. This will allow to distinguish between different processes and the complex interaction of climate, catchment characteristics (land use and land cover, geology, topography, shape…) and humans affecting stream chemistry.**

We re-phrased the research objectives in the introduction to clearly state what we are investigating:

L94-99: "We perform the analysis of magnitude, temporal trends, and seasonality of the in-stream concentrations with the goal of highlighting the differences across the eleven catchments and investigating the drivers of such differences. Specifically, we focus on the following research objectives: (i) investigating to which extent the solute concentrations are influenced by anthropic activities; (ii) exploring the dependence of solute concentrations on catchment characteristics; (iii) generalizing, if possible, the behaviors of selected solutes across different catchments by means of the slope in the C-Q relations."

Concerning hypothesis, being this a "data-driven" analysis and not a numerical or field experiment we prefer to refrain from formulating a-posteriori hypotheses, given also the challenge to test them with 11 catchments only.

**The influence of climate forcing on chemical variables was only vaguely discussed and not supported through any analysis, but it is required in the next version.**

We thank the Reviewer for this observation. In the manuscript there is not a specific analysis dedicated to the influence of climate on stream biogeochemistry, because the sample does not suit the requirements of size and independence necessary for a formal statistical analysis of climatic effects. The study is indeed based on only 11 catchments across Switzerland, partially nested catchments. Table 1 gives an overview of the catchments including mean annual rainfall, which could represent the major source of climatic variability across the catchments being other variables as temperature, humidity and solar radiation rather similar at catchment scale. However, the mean annual rainfall across the catchments vary in the range 1063 – 1506 mm/y; the only exceptions are Lümpenenbach (LU, 2127 mm/y) and Erlenbach (ER, 2182 mm/y) catchments, situated in the Alptal valley. These two catchments are also the two smallest catchments considered in the study, with an area at least three orders of magnitude smaller than the others and therefore some observed behaviors might be due to the wetter conditions but also due to the

smaller size. The effects of the two factors cannot be discerned with such a small sample size and any dedicated analysis to partition climatic effects is not meaningful given the similarity in climate of all the other basins.

**The number of figures in the main article and supplementary material is overwhelming. By refereeing to both main article and supplementary material, results in that the reader get lost and forgets about the main findings.**

We thank the reviewer for the comment and the reviewed manuscript has substantially different set of figures, which are outlined in the following.

**Figures contain lots of information, but the results are not equally explained for every figure. With regard to the figures, I suggest moving important figures into the main article and leaving "less" important ones in the supplementary material. To decrease the number of figures and focus on the main findings different figures could be combined, e.g. Figure 2 and 10, leave out Figure 3 (almost similar information as Figure 4?) and combine Figure 5 and 7.**

Figures have been modified in order to have more essential information. The number of figures was reduced and some figures were moved from the main article to the supplementary material and vice versa. Particularly, Figure 2 and 10 were merged (Figure 7), Figure 3 was deleted, Figure 5 was moved to the SI as well as Figure 7. Figure S1 and S4 were moved to the main manuscript. Concerning Figure S1, the pattern of DOC/TOC was removed, since it was not fundamental for the discussion.

Most of the Figures in the revised version of the manuscript show the background colors consistent to the topographic classes, so that the work results more consistent.

**In the presented work, a subset with consistent temporal data was used. However, it would be unfortunate to exclude how the stream chemistry changes in space. The full data set could be used to perform an additional spatial analysis to infer at which scale the effects of climatological forcing, catchment characteristics and human influence can be detected.**

The objective of our study is to analyze long-term water quality data in order to investigate the possible signature of anthropic activity and of catchment characteristics. The focus of this article is on purpose on the temporal analysis more than on the spatial variability and use time-series of a length that is deemed sufficient to filter short-term variability and obtain robust long-term behaviors. An exhaustive spatial analysis would require different criteria for station selection, different type of analysis, in other words a very different study with another database, which we consider to be out of the scope of our study. The eleven selected catchments, meaning water quality samples in eleven points across the entire Switzerland provide some information about spatial variability that is included in the narrative of the article, but are not sufficient for an extensive and statistically robust analysis of spatial variability.

**It is necessary to correlate the chemistry with other variables than agricultural land use as land cover, geology, urban area etc. To strengthen the findings it is necessary to perform a comparison between variables and catchments and test weather the findings of Figure 4, 6, 7 and 9 are significant different**

We applied the Spearman's non-parametric test (*Spearman*, 1904) to test the correlation between catchment characteristics and both the solutes concentrations and the catchment behaviors, like other studies in literature do (e.g., *Godsey et al.*, 2009; *Moatar at al.*, 2017). First, this test quantifies the correlation between the median concentration of each solute across the catchments and the variables, representing the different catchment characteristics. Consequently, we compute the correlation between the $b$ exponent, representing the biogeochemical response of the catchment and the same variables: catchment area, average altitude, mean annual precipitation, mean annual discharge, lake area in the catchment, percentage of catchment area in Swiss Plateau morphologic zone, percentage of catchment area in the Alpine morphologic zone, percentage of intensive agricultural area, percentage of extensive agricultural area and inhabitant density. Table 1a, Table 1b and Table 1c show the results in terms of Spearman's correlation coefficient for between the median concentrations of each solute, the low- flow $b$ exponent and high-flow $b$ exponent respectively with the catchment characteristics outlined above.

*Table S1.* Spearman's correlation coefficient between (a) the median concentrations of each solute, (b) the *b* exponent for low-flows and (c) the *b* exponent for high-flows across the catchments with some catchment characteristics listed in the first column. The green cells represent the significant correlations, i.e., the correlations characterized by a p value lower than the significance threshold α fixed at 0.05.

| (a) Median solute concentration | | | | | | | | | | | | |
|---|---|---|---|---|---|---|---|---|---|---|---|---|
| | $Ca^{2+}$ | $Mg^{2+}$ | $Na^+$ | $H_4SiO_4$ | $K^+$ | $Cl^-$ | $NO_3$ | TN | DRP | TP | DOC | TOC |
| Catchment area | 0.36 | -0.03 | 0.37 | 0.38 | -0.38 | 0.18 | 0.27 | 0.33 | 0.18 | 0.04 | 0.54 | 0.51 |
| Average altitude | -0.64 | -0.55 | -0.67 | -0.73 | -0.08 | -0.63 | -0.78 | -0.75 | -0.75 | -0.54 | -0.32 | -0.41 |
| Mean annual precipitation | -0.05 | 0.00 | -0.01 | -0.02 | 0.15 | 0.11 | 0.12 | 0.05 | 0.15 | 0.17 | -0.34 | -0.35 |
| Mean annual discharge | -0.69 | -0.54 | -0.65 | -0.72 | -0.28 | -0.57 | -0.66 | -0.67 | -0.58 | -0.51 | -0.33 | -0.43 |
| Lake area | 0.45 | -0.13 | 0.09 | 0.19 | -0.24 | 0.02 | 0.09 | 0.12 | -0.02 | -0.19 | 0.42 | 0.30 |
| % of Swiss Plateau area | 0.83 | 0.63 | 0.83 | 0.85 | 0.25 | 0.73 | 0.91 | 0.91 | 0.80 | 0.63 | 0.42 | 0.51 |
| % of Alpine area | -0.75 | -0.44 | -0.70 | -0.77 | 0.03 | -0.63 | -0.83 | -0.83 | -0.74 | -0.57 | -0.22 | -0.35 |
| % of intensive agricultural area | 0.88 | 0.38 | 0.78 | 0.87 | -0.15 | 0.68 | 0.85 | 0.87 | 0.78 | 0.58 | 0.54 | 0.63 |
| % of extensive agricultural area | -0.81 | -0.35 | -0.58 | -0.68 | -0.18 | -0.52 | -0.63 | -0.64 | -0.52 | -0.41 | -0.49 | -0.52 |
| Inhabitants density | 0.77 | 0.40 | 0.70 | 0.72 | 0.05 | 0.57 | 0.67 | 0.69 | 0.56 | 0.40 | 0.55 | 0.57 |

| (b) b exponent inf | | | | | | | | | | | | |
|---|---|---|---|---|---|---|---|---|---|---|---|---|
| | $Ca^{2+}$ | $Mg^{2+}$ | $Na^+$ | $H_4SiO_4$ | $K^+$ | $Cl^-$ | $NO_3$ | TN | DRP | TP | DOC | TOC |
| Catchment area | 0.51 | 0.54 | -0.14 | 0.17 | -0.09 | -0.21 | -0.28 | -0.17 | -0.20 | -0.14 | -0.22 | 0.15 |
| Average altitude | -0.72 | -0.63 | -0.24 | -0.40 | -0.16 | -0.46 | -0.49 | -0.43 | -0.26 | 0.24 | -0.25 | 0.33 |
| Mean annual precipitation | 0.01 | -0.10 | 0.04 | 0.15 | 0.16 | 0.42 | 0.58 | 0.44 | 0.58 | 0.48 | 0.34 | 0.07 |
| Mean annual discharge | -0.57 | -0.40 | 0.14 | -0.27 | 0.45 | 0.24 | 0.27 | 0.23 | 0.41 | 0.62 | 0.33 | 0.49 |
| Lake area | 0.55 | 0.58 | 0.08 | 0.18 | -0.04 | -0.08 | -0.40 | -0.25 | -0.28 | -0.25 | -0.28 | -0.11 |
| % of Swiss Plateau area | 0.68 | 0.49 | -0.03 | 0.49 | -0.18 | 0.09 | 0.10 | 0.06 | 0.18 | -0.19 | 0.22 | -0.16 |
| % of Alpine area | -0.56 | -0.32 | 0.17 | -0.32 | 0.25 | -0.08 | -0.05 | -0.02 | -0.11 | 0.34 | 0.02 | 0.38 |
| % of intensive agricultural area | 0.76 | 0.48 | -0.05 | 0.51 | -0.19 | 0.14 | 0.13 | 0.07 | 0.18 | -0.28 | 0.05 | -0.28 |
| % of extensive agricultural area | -0.75 | -0.48 | 0.08 | -0.52 | 0.44 | 0.12 | 0.10 | 0.13 | 0.24 | 0.44 | 0.45 | 0.45 |
| Inhabitants density | 0.70 | 0.53 | -0.13 | 0.48 | -0.38 | -0.13 | -0.07 | -0.20 | -0.19 | -0.37 | -0.17 | -0.18 |

| (c) b exponent sup | | | | | | | | | | | | |
|---|---|---|---|---|---|---|---|---|---|---|---|---|
| | $Ca^{2+}$ | $Mg^{2+}$ | $Na^+$ | $H_4SiO_4$ | $K^+$ | $Cl^-$ | $NO_3$ | TN | DRP | TP | DOC | TOC |
| Catchment area | 0.78 | 0.59 | 0.21 | 0.13 | -0.12 | -0.40 | -0.17 | 0.16 | -0.35 | 0.09 | 0.03 | 0.50 |
| Average altitude | -0.69 | -0.77 | -0.28 | -0.53 | -0.75 | -0.40 | -0.53 | -0.18 | -0.28 | 0.30 | -0.29 | -0.04 |
| Mean annual precipitation | -0.14 | 0.01 | -0.53 | 0.10 | 0.40 | 0.24 | 0.37 | 0.06 | 0.35 | -0.41 | 0.30 | -0.35 |
| Mean annual discharge | -0.62 | -0.63 | -0.65 | -0.43 | -0.25 | 0.01 | -0.06 | 0.03 | 0.05 | -0.18 | -0.02 | -0.38 |
| Lake area | 0.67 | 0.56 | 0.49 | 0.01 | -0.15 | -0.28 | -0.38 | 0.03 | -0.53 | -0.19 | -0.05 | 0.28 |
| % of Swiss Plateau area | 0.72 | 0.74 | 0.41 | 0.48 | 0.47 | 0.15 | 0.38 | 0.47 | 0.23 | -0.02 | 0.70 | 0.30 |
| % of Alpine area | -0.59 | -0.62 | -0.21 | -0.35 | -0.46 | -0.11 | -0.31 | 0.04 | -0.13 | 0.22 | -0.22 | -0.02 |
| % of intensive agricultural area | 0.79 | 0.75 | 0.27 | 0.47 | 0.42 | 0.00 | 0.24 | 0.36 | 0.13 | -0.09 | 0.59 | 0.29 |
| % of extensive agricultural area | -0.76 | -0.76 | -0.45 | -0.53 | -0.20 | 0.19 | 0.09 | 0.21 | 0.24 | -0.02 | 0.02 | -0.53 |
| Inhabitants density | 0.77 | 0.73 | 0.33 | 0.40 | 0.11 | -0.28 | -0.07 | 0.21 | -0.18 | 0.03 | 0.41 | 0.56 |

The green cells represent the statistically significant relations, i.e., the p-values of which are lower than the threshold of 0.05, above which the test is statistically significant. The correlation between the median concentrations and the catchment characteristics is significant in more cases than when correlating the *b* exponents, for both low-flows and high-flows. We added the tables in the SI (Table S1a, S1b and S1c) and commented in L267-276 concerning both the significant relation between the land use and the median nutrient concentrations in rivers and our original concerns about the significance of the test on a sample of 11 catchments only.

**In addition, to be able to judge the validity of the results and limitations of the dataset, it is necessary to consider measurement errors and detection limits of chemical variables.**

We agree with the reviewer that the manuscript was lacking comments concerning measurement errors and detection limits of chemical values. Concerning the detection limits, we considered them during the preliminary data elaborations, when the database was pre-processed to clean outliers due to issues in detection limits. We added the following statement referring to a very recent paper published on the NADUF dataset where more extensive information about data collection is reported:

L:126-129: "Data are collected following ISO/EN conform methods for water analysis and all the data are subjected to an extensive quality control as described in *Zobrist et al.*, 2018. However, we additionally took into account possible errors deriving from fixed detection limits deleting the values below the detection threshold."

**The discussion and conclusion sections are wordy, without highlighting the new findings.**

In the revised manuscript the discussion and conclusion sections are more essential and linear, focused on the research objectives. The new findings were highlighted, especially in the conclusion (L582-607). But please see also changes in the discussion section (L 444-446, L457-458, L508-516).

**To explain the data, different streamflow generation and runoff processes were hypothesized to occur. However, the discussion section lacks debating the temporal sampling interval and its ability describe potential occurring processes at shorter time scales.**

This is an important observation and we agree that the discussion should include statements on the ability of the temporal sampling interval to describe processes potentially occurring at the shorter time scale. The sampling method of the NADUF database is flow-proportional and it is characterized by a 14-days frequency. The low sampling frequency does not allow recognizing short-time scale signals in the biogeochemical signature of rivers, even though these are integrated in the 14-days frequency, being the sampling "flow-proportional".

Water quality programs established in many catchments around the world record data at quarterly, monthly, 14-days or weekly frequency because of time and financial constraints. Despite this low sampling frequency, we think that these data have a high information content if used in the proper way. The representation of short time scales processes cannot be the goal of this kind of database, but the description of long-term patterns of water quality parameters in relation to external forcings (i.e., anthropic activities and catchment characteristics) can. We pointed out the low-frequency of our dataset throughout the manuscript, both in Section 2 and in Section 5.3: (L122-125 and L510-516).

**In addition, it is necessary to refer to the current understanding of streamflow generation, runoff processes and stream network connectivity observed around the world. But also studies from Switzerland, which is known as a country to have a long, history and good knowledge of detailed small-scale catchment processes understanding (from lowlands to Alpine regions), needs to be included. The authors need to revise this section and including these points to be able to put findings into a spatial and temporal perspective.**

We thank the reviewer for the suggestion and we enriched the discussion with examples from literature focused on the hydrological processes understanding whenever it was suited.

**The manuscript will benefit by streamlining the discussion and conclusion with focus on the effects of climatological forcing, catchment characteristics and human influence on the spatial and temporal variability of chemical variables.**

Concerning climatological forcing effects and spatial variability please refer to our reply above. The discussion and conclusion sections were modified to streamline the research objectives of the work.

**It would be also valuable to add a discussion section, from a hydrologist-scientific point of view with an outlook for potential next steps or a critical note on the data collection and what could / should be done differently to address certain future questions.**

We thank the reviewer for these suggestions. Since the data issue is widely discussed in water quality studies (e.g., *Stelzer and Likens*, 2006) we just added a short note on data collection, reported in the previous reply.

A brief outlook with potential next steps was also included in the conclusion (L608-611).

**Specific comments referring to line**

**27**             "*Both natural and anthropogenic factors impact...* **"If this is the case, it is necessary to perform an appropriate analysis to separate the different factors.**

See the comment above concerning the hypothesis. In our data-driven study we detect differences across the catchments in terms of concentration patterns and we highlight the relation with natural or anthropogenic drivers. However, we now stated in a clearer way what we are investigating with each analysis throughout the manuscript.

**36, 52**          … r*esidence time…"* **residence times are usually calculated using conservative tracer which is not part of this study. Please clarify.**

As the reviewer correctly points out, residence times are usually calculated using conservative tracers and this is not included in our study. However, we mention residence time in the introduction, while framing the problem. The concentrations observed in the rivers are the result of different factors, among which also residence time. As we state in line 53, the solute release reflects the mixing processes which take place in the catchment and which are highly dependent on residence time. We are persuaded that mentioning residence time while framing the problem is appropriate as long as we do not state that it is the focus of our study, because it is a component that highly contributes to the observations we analyze. We also mention residence time in the discussion section of the revised version.

**51**          **If the catchment structure is important did you test for this?**

We thank the reviewer for the comment. The term "catchment structure" is too general and not appropriate for our study. We modified "catchment structure" into "catchment geomorphology and land use", which are more specific and linked to our work.

**94**          **Please clarify the focus of the study, is it human or anthropogenic. How do you distinguish between human and non-human influences?**

We stated clearly the definition of the term "anthropic activities", which is now used in a coherent way throughout the manuscript. Throughout the manuscript we refer the adjective "anthropic" to the terms "activities", "pressure", "presence" and "origin", while the adjective "anthropogenic" to the terms "factors", "signature" and "disturbances": Indeed, while the first is used as synonym of "human", the second is used as synonym of "of human origin". Please refer to the comment above.

**100**          **In the table, please indicate the start and end of the available data of the different catchments. What is the impact of the length of different time series? It could be that different decades where drier compared to others. This could affect the results. Please test and explain.**

We modified Table 1 adding the period of time spanned by the time series and the number of consecutive years taken into account in the study. The length of different time series could potentially have an impact on the results. Figure 1 shows the mean monthly discharge across the decades for each catchment with at least 20 years of data. The black line refers to the period 2005-2015, which corresponds to the period of shortest time series available (ER and LU).

[Figure]

*Figure 1: Mean monthly discharge across the decades for each catchment with at least 20 years of measurements.*

The discharge, as well as the external forcing, which affect the concentrations in rivers (e.g., number of inhabitants, pollution sources, intensive agricultural surface), do change in time. In our study we accounted for this aspect by investigating the trends of concentrations in time and by analyzing the pattern of C-Q relations across decades. Additionally, starting from the consideration that the mean discharge, at the station Rhone – CH, has decreased in time, we looked at the mean monthly average concentration of all the solutes for the entire monitoring period. If we hypothesize that a change in discharge is strongly impacting the in-stream concentrations, we may expect to observe a trend or a change in all solutes. Figure 2 shows the mean monthly trend coloring differently concentrations from the first to the last year of the monitoring period for the CH station. Only $Na^{2+}$, $Cl^-$, DRP, $NO_3$, total nitrogen and total phosphorus follow a clear monotonic trend, but our study argue that these trends are mainly due to external forcing. $Ca^{2+}$, for example, is related to rocks weathering and while wetter or drier conditions could influence its dynamic, we do not see this. Indeed, despite the trend toward decreasing discharge, $Ca^{2+}$ concentration of the Rhone river does not show any trend. Therefore, while we explicitly analyze trends in concentrations, we do not think that considering time series of different lengths has a strong impact on results, as can be seen for the Rhone river. However, we acknowledge this potential issue in the manuscript L199-203.

[Figure]

*Figure 2: Long-term solutes trends. Each line represents the monthly average concentration of each solute. The color bar indicates the years of the monitoring period, from the first year (blue) to the last year (red). The presented figure refers to the Rhone – PO catchment.*

**102-103     Please rephrase "basins" and "*large catchments*". Especially since ER LU are rather small catchments.**

The adjective "main" is not referred to Switzerland but it is relative to the case studies we considered. We meant that the study addresses 11 case studies on 6 rivers, because some of them are sub-catchments in the same river basin. In order to avoid any possible misunderstanding we rephrased the sentence (L106-107).

**109     Here you could highlight the long-term character of the data.**

We thank the reviewer for the suggestion. We added the following statement: "However, the dataset is especially suitable for the investigation of long-term trends, thanks to the length of the time series, which spans from 11 to 42 years (Table 1)" at lines 125-126.

**116-119     This section is general. Please be more specific and refer to figures.**

We rephrased and extended significantly the entire study sites description. Please refer to the reply further above, where the description is reported.

**117     Please explain what "anthropogenic pressure" is.**

Please, refer to the comment above about the rephrasing of lines 138-142.

**120     In the discussion section please discuss how the temporal sampling could affect your results. If these samples are long stored, could reactive processes alter the sample and affect the results? Please comment on this.**

The samples are collected flow-proportional and analyzed every two weeks. The process of collection, storage and conservation is conducted following the standard procedures. Collected samples are stored at 4°C and transported in cooled containers to the laboratory for the analyses. In the revised version of the paper we refer to the very

recent publication *Zobrist et al.* (2018), where the procedure of the NADUF data collection is explained exhaustively.

Moreover, we are aware that a sampling frequency of two weeks does not allow to detect short-scale processes and the integral measurement over the 14 days period averages out the concentration and discharge patterns. We comment on this at L122-129.

We refer to the limitations due to the low sampling frequency also in the discussion section and in the conclusion.

**130,222,250  Phosphor but also other variables could be also due to weathering. Please comment on this.**

We agree with the reviewer that not only the mentioned solutes are generated by weathering. We prefer classifying nitrogen, phosphorus and organic carbon species separately from the geogenic solutes because, from literature, it is well known that they interact differently with microbes and vegetation from ions like calcium or magnesium. Indeed, they are nutrients for biotic activities and they are fundamental components of the soil biogeochemical cycles, while the solutes we classified as 'geogenic' are less involved in the biological cycles at least in comparison to the quantities that are produced by weathering.

**134        Why are these variables available and not shown or used in the analysis?**

In our analysis we wanted to focus on direct measurements of solutes more than on parameters representing indirectly the concentrations of the different solutes.

**147        Please explain why this classification was used and not a classification based on the regime type, precipitation, geology, or land use.**

We agree with the Reviewer that the previous version of the manuscript was lacking an explanation of our choice, we therefore integrated this explanation al lines 174-183.

**148        Is there a different term for hybrid catchments? It sounds like a mythological creature.**

We long thought about this name, and we concluded that it was the best descriptive adjective for these catchments. In other words, we could not find a better alternative definition to that class of catchments. Any better suggestion would be very much appreciated.

**149-152        This sentence is a conclusion. Please move in the appropriate section.**

We thank the reviewer for the suggestion and we moved the statement to section 4.2, L282-284. The statement was also rephrased.

**203        Please explain these ratios.**

We removed the DOC/TOC ratio because it was not a fundamental analysis for our study, both in the text and in Figure S1 (Figure 2 in the new version). We also added the explanation of the ratios and rephrased the sentence (L240-247).

**219        Whys is this not surprising? Please explain better or rephrase the sentence.**

We rephrased the sentence here L260-261: "DOC and TOC concentrations are very high in Lümpenenbach (LU) and Erlenbach (ER) catchments (Figure 3c), which are the smallest catchments with the highest average yearly precipitation rate and very low anthropic presence." and we explained the hydrological processes behind this result in section 5.2 (L483-495).

**230        "…*seasonality of streamflow Swiss Plateau catchments is determined by a combination of precipitation and snowmelt.*" Isn't this the case in pre and or Alpine catchments? Please explain.**

In general this is the case, but snow and glacier components are much more prominent in alpine catchments. According to *Weingartner and Aschwanden* (1992), the regime in the Swiss Plateau is nivo-pluvial, while in the Alpine area is glacial or nivo-glacial.

**249**     **" … there are factors…" which factors please specify.**

These factors are explained in the Discussion section, so we did not explain them here but rephrased as: "Type B solute ($Mg^{2+}$, TP, DOC and TOC) response shows a higher variability index in Alpine catchments compared to types A and C, thus indicating that, among the factors controlling the seasonality of biogeochemical response, there are factors that are specific to the Alpine environment, which are outlined in Section 5.2.*"*

**307**     ***"Solute export across catchments seems to be mostly controlled by anthropogenic factors rather than by catchment characteristics."* None of the analysis supports this conclusion! Please perform an analysis, which separates the effect of climatology, geology and human.**

We rephrased the statement as "The cause-effect relation between the observed in-stream concentrations and the anthropic activities is sometimes evident in the concentration magnitude, seasonality and long-term trends". Actually, we are not able to compare the strength of the impact of catchment characteristics and of anthropogenic activities, since isolating each catchment characteristic and relating it to stream signatures is not straightforward, a part from some special cases which are evident and discussed in the manuscript. In this revised version of the manuscript we made clear this point. Concerning the climatology, please refer to the specific reply above.

**313-314**     **How representative is such a ratio for agriculture seen ratio of the plateau catchments and e.g. LU is similar? Please comment on this.**

We thank the reviewer for the question. We found an inconsistency in the old Figure S1 (Figure 2 in the revised version) and we apologize for this. The ratios in the previous version of the manuscript were computed on the entire monitoring period of each station, but, since nitrogen and especially phosphorus recorded significant decrease of magnitude in time, the comparison referred to different time windows is not completely consistent. We therefore computed the ratio over the period 2005-2015, since it is covered by all the measuring stations. Figure 3 shows the comparison between the ratio as it was on the previous version of the manuscript (dotted line) and as it is in the reviewed version (solid line). Moreover, as we explained previously, the DOC/TOC ratio is not presented since it does not represent a fundamental information. We added the background colors referred to the morphologic zones. In the new version $NO_3/TN$ in LU and ER are significantly lower than in human-impacted

[Figure]

*Figure 3: NO3/TN (green) and DRP/TP (red) ratio across the eleven catchments. The solid line represents the value of the ratios over the period 2005-2015, while the dotted line over the entire monitoring period of each station.*

catchments and this is consistent with the article narrative.

**316**     **This statement needs to be better explained. What are the differences and which other analysis were used?**

Because Figure 2 (former Figure S1) changed, also the relevant comment about the observed results changed. We rephrased the entire statement: L377-381: "The variability of the ratio between average $NO_3$ and TN concentrations across the different catchments, is comparable with that estimated by *Zobrist and Reichert* (2006), who observed

a variation from 55% in Alpine rivers to 90% for rivers in the Swiss Plateau. Both $NO_3$ to TN and DRP to TP ratios show a decreasing trend from more to less anthropic-impacted catchments, however the range of variability is higher for phosphorus species (from about 0.6 in Thur river to about 0.2 in Inn River)."

**319-320**          **Please provide a back of the envelope calculation to support this statement.**

The comment was speculative and based on Figure S1 of the previous version of the manuscript. Since we found out the inconsistency in Figure 2 (former Figure S1) we deleted the comment because it does not match the observations.

**324 -327**        **Consider moving Figure S1 into the main document to show the signal. Please explain the definition low intensive (one cow) vs. high intensive (several cows?). In winter cattle is kept indoors while in summer outdoors. Can you observe this in your data? If processes overlap, how is it possible to distinguish from dominant processes and natural vs. human processes? Please comment on this.**

We thank the Reviewer for the suggestion of moving Figure S2 to the main manuscript and we did it. In the revised version Figure S1 has become Figure 2.
The term "lower intensive agriculture" refer to the general concept of intensive and extensive agriculture, which is usually related to the fertilizers input introduced for yield production purposes. As the updated Figure 1 shows, most of the agricultural activities are concentrated in the north of Switzerland, while the Alpine region in the south is not suitable for intensive agriculture. We did not focus specifically on cattles or livestocks and we did not investigate whether a relation exist between cattle and the observed in-stream concentrations as the information required for this analysis is to a large extent not available in a form that is consistent with the solute data. Please consider that catchments are typically large enough to host a quite diverse range of agricultural activities.
About the overlapping of processes, the point we want to make is that with the analysis of the seasonality we could isolate the impact of streamflow seasonality (natural forcing) and observed seasonality of concentrations. If the latter does not follow the seasonality of discharge other factors might contribute to this dynamics, and we discussed which factors are the most plausible.

**L330-331**        **Please clearly state the difference between agriculture and human activities. Also, in terms of the type of signal one could expect.**

We added the definition of anthropic (or human) activities in L137-141. Agriculture is one of the anthropic activities and this is the reason why we write "through agricultural practices or by means of **other** human activities". The other human activities might be those mentioned in L138-142.

**333-338**        **This section is not clear. Please guide the reader through Figure S2. In which way does the coefficient de Pardé of discharge and loads relate to each other? Please explain.**

We explained more extensively the link with Figure S2 (Figure S3 in the revised version) and rephrased accordingly (L397-400).

**354**            **Throughout the document it is necessary to clearly state what are areal sources e.g. cities, point sources e.g. Water treatment plants, line sources e.g. streets and what the implications are on the chemical signal. This because data at the catchment outlet might not be representative for all of the catchment. Please discuss this critically.**

In the revised version in the Section 2 we describe the possible sources of pollution while describing the study site (L138-142) and we represent them in Figure 1. Because we did not explicitly write that spreading the de-icing salt on roads is an anthropic activity which might have an impact on the in-stream signature, we clarify this at L259-262.

**363**            **If this figure is important and discussed, it should be in the main manuscript. It would be also interesting to add a "natural" catchment to show the difference between pristine and human influence catchment.**

We thank the reviewer for the suggestion and we moved the Figure S4 to the main manuscript as Figure 6. Moreover, we added the comparison between a "natural" and a human-impacted catchment and modified the text accordingly.

**367-370** **This sentence is not clear, what are secondary effects. Could a Ca/Mg ratio be useful to show whether the effect is really from fertilizers and not from weathering? It also is necessary to provide data on what would be a natural background value and what could additionally come from fertilizers.**

We thank the reviewer for this observation because we misinterpret what *Zobrist* (2010) claims and we have the opportunity to clarify and fix the manuscript. *Zobrist* (2010) analyzes the NADUF long-term patterns of the Alpine catchments. A trend analysis is also conducted in his work and, specifically regarding magnesium, a non-monotonic trend is observed over the period 1975-2006, as we observed. *Zobrist* (2010) hypothesizes that it might be due to the release of magnesium from soils, which cumulated it through fertilizers application, since many fertilizers contain magnesium as a minor ingredient. However, referring to Rhine and Rhone river, they observe that calcium, over the same period, had the opposite non-monotonic trend, even though with a much slighter magnitude. Therefore, their conclusion is that the most influencing factor might be the temperature of water, which in Rhone and Rhine has been increasing significantly in 1975-2006. We agree with *Zobrist* (2010) concerning the catchments where the non-monotonic trends of $Ca^{2+}$ and $Mg^{2+}$ exist and are opposite to each other (i.e. first increasing then decreasing for $Mg^{2+}$ and first decreasing then increasing for $Ca^{2+}$), which are Rhone and Rhine (Figure 4). For the Aare catchment, instead, which is located in the Swiss Plateau region, only magnesium shows a non-monotonic trend, while calcium does not show any trend (Figure 4). In this case the hypothesis of temperature as main driver does not find a confirmation. We therefore consider the hypothesis that it might be an effect of the fertilization, also because Aare (i.e., BR and HA) is mainly an agricultural catchment (Figure 1). We modified the text in L420-431.

We further discuss the comparison with the background concentration in L358-362: "Phosphorus and nitrogen are the main nutrients applied for agricultural fertilization and, Figure 3d shows a decreasing pattern from mostly intensive agricultural catchments to forested catchments. Indeed, taking the concentrations of $NO_3$ and DRP registered at ER as reference background concentrations [*Zobrist*, 2010], the concentrations in all the other catchments are significantly higher."

[Figure]

*Figure 4: Mg/Ca ratio and normalized Mg and Ca pattern over the monitoring period. The left upper and bottom panels refer to Aare river (BR), while the right upper and bottom panels refer to Rhine river (WM). The upper panels represent the ratio Mg/Ca computed with the average monthly values, while the bottom panel represent the mean monthly concentration of Mg (blue) and Ca (red) normalized by their respective average value over the entire monitoring period. The black horizontal line corresponds to the value 1.*

**378** **This seems a hypothesis and not supported by your analysis.**

We rephrased accordingly to the reply above (L444-446): "A statistically robust link between catchment characteristics and river biogeochemical signatures is not straightforward, since spatial heterogeneity in river

catchments and the limited sample size, make the search for cause-effect relations between catchment characteristics and in-stream concentrations challenging."

**379 386**    **By including a geology map and other catchment characteristics will reveal which catchments are affected and by which percentage! Please add and comment.**

The description of catchments has been widely revised. Please, refer to the relevant comment above.

**387**    **This sentence is not clear, phrasing and hydrological processes. Atmospheric forcing causes also a temporal variability in steep and flat catchments. Please comment / explain.**

In our analysis of seasonality we isolated the effect of the topography dividing the catchments into topographic classes, which are characterized by different hydrological regimes. Atmospheric forcing also causes temporal variability in steep catchments, but we cannot discern the impact of the various climatic forcing because our dataset is quite homogeneous in this regard. We rephrased with a more exhaustive explanation: L457-460.

**389**    **Also in bare fields, soil erosion can be high. Can you see this in the data?**

We agree with the Reviewer that erosion can be high also on bare fields, but the erosion due to Alpine steep morphology is much more significant than the one characterizing bare fields in lowland catchments (which are extremely rare in Switzerland), as the boxplot showing suspended solids concentrations in Figure S5 demonstrates.

[Figure]

***Figure S5.*** Suspended solids concentrations across the catchments. The boxplot shows the suspended solid concentrations across the eleven catchments, ordered from the Swiss Plateau (blue background), to the hybrid (light blue background) and Alpine (yellow background) catchments. The Alpine catchments show much higher concentrations and variability.

This Figure was integrated in the SI of the revised version of the manuscript and we explained better this point in the text (L460-461).

**396-414**    **This section is not clear. In the "alpine" categorized catchments, not every catchment has a glacier. Please comment and explain. This is where global and local runoff processes understanding need to be added.**

Among the catchments classified as "Alpine" in our study Rhone, Rhine and Inn include glaciers in their domain (e.g., *Huss and Fischer*, 2016), while the smaller catchments ER and LU do not. We rephrased and added literature related to hydrological processes (L465-482).

**420-426**    **I believe that lakes dampen the signal, but where is the analysis to support this statement? In addition, how do other processes, e.g. instream processes, change the signal of interest?**

A large body of literature reports the dampening of biogeochemical signals of lakes (e.g., *Ito et al.*, 2007, *Kaste et al.*, 2003; *Wurtsbaugh et al.*, 2005). However, since we do not provide any analysis proving this statement and it is not fundamental for the scope of our work, we removed the statement concerning the influence of lakes. Also different in-stream processes contribute to the signal modification, e.g., biological processes and chemical reactions such as solute precipitation/dissolution and sorption/desorption. All these processes are implicitly accounted for by the observed concentrations and additionally filtered by the flow-proportional 14-days resolution data. This is acknowledged in the revised version of the manuscript L122-125.

**396-414,438-443**      **Including recent findings on streamflow generation and runoff processes (e.g. America, Europe and Japan) might help to explain the different signals and improves the discussion.**

We thank the reviewer for the suggestion and we integrated references related to streamflow generation and runoff processes. Since the entire sections 5.2 and 5.3 were deeply revised, please refer to the text.

**476**      **Please define what low-flow is. Does this differ among catchments and how does it affect concentrations. Please comment.**

Low-flow was mentioned at former L476 (now L567) and refers to low-flow conditions defined in L214 "$q_{50}$ to separate flow below the median (low-flows) and flows above the median (high-flows)." and also in L506-508: "In fact for low-flow conditions (i.e. q<$q_{50}$) this is typically associated with biogeochemical processes of solute removal, while for high-flow conditions (i.e. q>$q_{50}$) it is generally associated with the capacity of the flow to entrain particles containing the solute".

The threshold defining low-flow and high-flow conditions is computed as the median daily discharge $q_{50}$ and it varies from catchment to catchment. In the revised version of the manuscript we comment on this choice at L513-516.

**496**      **Why are the atmospheric forcing and catchment characteristics less evident? Based on what? Please explain and rephrase.**

We rephrased accordingly to the replies above: "The analysis of magnitude, seasonality, and temporal trends revealed clear cause-effect relation between human activities and solute concentrations, while the detection of the influence of catchment characteristics is less straightforward and can be only captured in a quantitative but not statistically significant way due to the spatial heterogeneity of catchment characteristics and the relatively small sample size (11 catchments)."

**499**      **Despite this spatially rich dataset, how many samples e.g. catchments would one need. What is the effect of scaling in your signal and how would this effect your results - macro pattern vs. micro pattern?**

Our dataset is spatially rich in the sense that it covers basically the entire Switzerland, but most of the catchments are very big and therefore they are non-homogeneous in terms of spatial characteristics. To detect a stronger influence of catchment characteristics in our analysis we would need a larger sample not only in terms of number of catchments but also in terms of distinct spatial characteristics. Currently, each catchment includes a variety of spatial characteristics and finding a cause-effect relation between one specific characteristic and the signature in the in-stream water is not possible. Therefore, it is not only a matter of number of catchments that are needed but also how catchment characteristics vary across catchments.

**Table 1**      **In the caption there is written north-south gradient. Is this gradient geographically or rank in the table? Please clarify. Maybe change in the header row the names to - mean annual precipitation and mean annual discharge?**

Since we modify Figure 1 showing the north-south gradient and we explain better in the text (L160-161), we remove this information here in Table 1 to avoid redundancy. We also changed the header of Table 1 as suggested.

**Figure 1**      **To better link the different catchments use a consistent color scheme as in figure 6.**
                **Please add basic information such as scale bar, north and legend. Also, add information on land cover and land use and large cities, geology and country names. It is difficult to see where lakes are and why streams, crossing the Swiss borders , are not represented.**

We thank the reviewer for these suggestions and we modified Figure 1 adding information which helped in explaining better the case studies in Section 2. The new figure and the respective caption are:

[Figure]

*Figure 5:* Map of NADUF monitoring stations and description of the study sites. The upper panel represents the study sites. On the left the Swiss Plateau (blue) and the Alpine catchments (yellow), on the right the hybrid catchments (light blue). The bottom panel describes the study sites in terms of topographic areas (left), land cover (center) and anthropic pressure (right).

We did not include the geology map, but in the text we explain the geologic zones, which are strictly related to the Swiss regions, shown in the bottom panel (left) of Figure 1. This information is sufficient for our analysis, while a geological map would not be straightforward to understand in the context of this article and would provide additional information, which is not required for our study.

**Figure 2      To help the reader please specify in caption what b and q50 means.**

We merged Figure 2 and Figure 10 as suggested by the reviewer and the resulting figure appears in the revised version as Figure 7. In the relevant caption we added the definition of $b$ and $q_{50}$:

**Figure 3      Why do some variables have species in their name and other not? Maybe add also the CV to compare different chemical variables. Due to similarities with figure 4, this figure could be removed.**

We removed this Figure as suggested by the reviewer.

**Figure 4      Give boxplots same color of catchment classification used in figure 6. Which catchment is WM? In Figure 1, this catchment is not visible and might be a typo? In d) one could argue on an existing decreasing trend with decreasing percentage of agriculture. This due to e.g. outliers e.g. BR-HA where after the information seems to flatten out. Please provide statistical test and in addition, a significant test to compare the different catchments and support your statement. In addition, is there a real difference in the light of measurement accuracy? Generally, are these concentrations and their variability high or low? Maybe compare the concentrations with observations elsewhere and report in the discussion.**

We thank the reviewer for the suggestion. We modified Figure 4 (Now Figure 3) using colors according to the topographic classes. WM and VW are the same catchment but the station in the original database changed the name during the monitoring period. The error originates from this inconsistency; we now used a single name throughout the manuscript so that it is going to be consistent.

In Figure 3d the statistical Mann-Kendall modified test reveals that no trend is statistically significant for each of the solutes. However, we are considering only eleven catchments, thus testing a trend across eleven points, which is challenging per se. However, a remarkable difference can be observed between agricultural dominated and forest dominated catchments, so that we think the figure is appropriate for the manuscript.

We discuss the measurement accuracy and refer to *Zobrist et al.*, 2018 for further information and added comments about the variation compared to background concentrations in L369-373.

**Figure 5** **Why the different streams were clustered according to a hydrological regimes classification? Do the different regimes really cluster e.g. only Inn and Rhine seem to be similar regimes but others not. Please explain. Caption: "*Each point represents the monthly average discharge …*" should not this be a ratio as described in the y-axis? "*Hybrid catchment*" Maybe choose a different term.**

We justify the basin clusters according to their hydrological regime in L174-179.

Former Figure 4 was moved to the SI because it is not fundamental for results, but shows the discharge seasonality. Former Figure 5 is Figure 4 in the revised version.

We thank the reviewer for highlighting the error in the caption and we fixed it.

Concerning the definition "hybrid catchment", please refer to the specific reply above.

**Figure 6** **Please be consistent with the terms. Is it index of variability or variability index? "*…discharge variability per catchment.*" This should be per catchment class. Rank the catchments as classes similar to Figure 5. Next to referring to section 3.2, please explain shortly the patterns and add a legend.**

We fixed the inconsistency in the wording. We used consistently the definition "Index of variability" throughout the manuscript and we changed therefore the x-axes label of Figure 6 (now Figure 4).
We adjusted the caption specifying that the index refers to the catchment class (same classes as ex Figure 5, now Figure 4) and explains the A, B and C patterns.

**Figure 7, 262** **Please better explain in the result section the signals and processes. The y-axis should be labeled as mean monthly concentration. Why was only the station at Rekingen shown? It would be interesting to see other catchments Alpine vs Urban influenced catchments and compare the change in amplitudes or signals.**

Figure 7 in the manuscript shows which kind of analysis has led to the observations of Figure 8. It is not widely discussed because the focus of the trend analysis is on Figure 8 (Figure 5 in the new version), which is a summary of the patterns we showed in Figure 7 and that were computed for every catchment. Given the marginal role of Figure 7 we moved it into the supplementary information, as Figure S1.
The change of amplitude of concentration in agricultural or urban catchments, although in terms of average variation for topographic class, is widely discussed in the seasonality analysis and it is out of scope for the trend analysis.

**Figure 9** **Please add labels to each plot a-d and specify which line is which catchment. Right panels) Why was only the station at Rekingen shown? It would be interesting to see other catchments Alpine vs Urban influenced catchments and compare signals.**

We modified Figure 9 (Figure 7 in the revised version of the manuscript) as follows:
-   Labels a-d were added to left panel,
-   Each line of the left panel is explicitly referred to a catchment,
-   C-Q relations are computed with specific discharge q in mm/h.
-   C-Q relations across decades are not only shown for one catchment, but one Alpine (PO) and one more anthropic (BR) catchment are compared.

**Figure S1** **The labels and captions are not coherent and make it difficult to understand. Please change and double check all figures.**

Figure S1 (Figure 2 in the new version) was modified following the suggestion of the Reviewer. We apologize for the inconsistency. DOC/TOC pattern was removed because the information was not fundamental for the discussion. We found an inconsistency in the previous version of the Figure, which we explained previously. We double checked the caption and plotted data and we added the background colors according to the topographic classes defined in Section 2. Also the catchment "VW" was turned into "WM", as explained in our reply further above.

[Figure]

***Figure 2.*** Ratios of DRP/TP (red), NO₃/TN (green) across catchments computed on the period 2005-2015. Both the patterns show a decreasing trend from more to less anthropogenically affected catchments (left-to-right of x axes). This pattern is more evident for phosphorus. Background colors refer to the morphologic classification explained in Session 3.1.

**Figure S2** …" *Monthly average of discharge*…" **The caption and y-label are not coherent and confuse the reader.**

We modified the y-axis label of Figure S2 to be coherent.

[Figure]

[Figure]

***Figure S2.*** Monthly average of discharge (black) and solute load (blue) normalized with the average of the entire monitoring period. The red horizontal line represents the vale 1 (i.e., mean). The subpanel (a) refers to Calcium and subpanel (b) to Nitrate. Calcium is originated by rock weathering and it largely follows the seasonality of discharge. Nitrate, instead, is related to the anthropogenic activities and in several catchments (i.e. Thur, Aare – Brugg) the load has its own seasonality, which is different from the seasonality of discharge.

**Several recent papers studied long-term series of water quality and discharge aiming to generalize behaviors of selected solutes across catchments in order to infer anthropogenic and catchment characteristic influences. This study provides some more results on Alpine streams. The authors analyzed geogenic solutes, chloride, nitrogen, phosphorus and organic carbon species, monitored by the Swiss National River and Survey Program for 11 Swiss rivers with a temporal resolution of 14 days as composite sampling (sampling represent an integration of the preceding 14 days) for more than 10 years. The analysis of basic statistics, seasonality, temporal trends and concentration-discharge behavior revealed impacts of human activities for some catchments. However, the influence of catchment characteristics is much less evident. This is probably due to the small number of analyzed catchments and to their area range which is very bi-modal (one group with catchment area around 5 000 to 30 000 km2 vs. 2 small catchments with area < 1km2) which do not help having a more quantitative spatial analysis.**

We agree with the reviewer that the sample of catchments is not large enough for an exhausting representation of the spatial variability and variety of Swiss river biogeochemistry. Indeed, the main focus of our analysis is on the long-term temporal trends. This has been clearly stated in the new manuscript.

**The manuscript needs to better explain the relation between temporal metrics and spatial characteristics.**

The manuscript was modified to include a clearer presentation of the spatial characteristics of catchments. A more exhaustive description was integrated in Section 2 and also Figure 1 was modified so that it can be more effective in illustrating the spatial characteristics of the catchments (e.g., urban areas, land use, topographic characteristics), see also L138-161.

**Another way of analyzing the results could be to consider the variation of these relationships along nested catchments (Rhein, Rhone, Aare).**

We thank the reviewer for the suggestion. This would be an extremely good idea with a different sample size, but, unfortunately, the sample of nested catchment is limited to 3. Therefore, we are skeptical about the possibility to obtain any robust pattern or generalization originated by these three nested catchments. An exhaustive spatial analysis would require different criteria for station selection, different type of analysis, in other words a very different study with another database, which we consider to be out of the scope of our study. Here we simply selected the catchments with at least 10 consecutive years of measurements for the investigation of long-term trends, as clearly indicated by the improved text of the revised manuscript.

**The manuscript has a relatively good structure, but the results could be presented in a more factual way, in order to better distinguish them from the discussion.**

We thank the reviewer for the positive comment and for the constructive criticism. We deeply modified the discussion section and the conclusion such that:
- We streamlined the discussion section focusing on the main research objectives;
- We eliminated from figures the unnecessary or redundant information;
- We integrated in the discussion section references to hydrological processed that are linked with the observations;
- We highlighted the new findings and the novelty of our study in both the discussion and conclusion sections.

**The conclusion needs to highlight the new findings of this work.**

We thank the reviewer for this comment and we now state more explicitly in the conclusion the key novelty of our study and the new findings (L586-590, L596-597 and L605-611).

**Specific comments:**

**Database and study sites**
**The authors do not present very well the database (numbers of data/years for each site and element, screening, discussion about the difference between composite sampling and grab sampling, representativeness of metrics calculated from composite sampling, especially for small catchments).**

We described more in detail than in the original manuscript the database by adding the time period and the number of consecutive years. We also added a comment on the sampling frequency and on the composite sampling in L122-129. Please note that the sampling is flow-proportional so even if we cannot capture short-term dynamics directly, these are proportionally weighted in the two-week average.

**It is not clear either whether all the calculated temporal metrics are based on mean bi-monthly concentration and discharge data time series. If this is the case, the authors need to discuss how this sampling design impacts the analysis of the temporal metrics (especially concentration-discharge relationships).**

All the calculated temporal metrics are based on mean bi-monthly concentration and discharge. However, please note that concentration is a "flow-averaged quantity" and not a snapshot every two weeks. The only statistic computed from hourly discharge data is the median daily discharge used in the C-Q relations and this is pointed out at lines 214-217. In the revised version of the manuscript, we discuss the impacts of the sampling design in L122-129. Low-frequency sampling can be one of the reasons why only 29% of the catchment-solute combinations have different behaviors between low-flow conditions and high-flow conditions. A higher sampling frequency would probably allow the detection of short-term processes (e.g. biological in-stream processes), which might differentiate more the behaviors across flow conditions. We added this point in the discussion section (L510-516).

**Catchment characteristics are not very well presented. Figure 1 could be reworked to present land use/land cover. Colors for catchment could be replaced by contour lines.**

We reworked Figure 1 to show with improved clarity the catchments we analyze, thus showing them on two different maps with colors, which are consistent with the topographic subdivision. Moreover, the morphological zones, land use and main anthropic areas are shown in an additional panel of Figure 1.

[Figure]

*Figure 6:* Map of NADUF monitoring stations and description of the study sites. The upper panel represents the study sites. On the left the Swiss Plateau (blue) and the Alpine catchments (yellow), on the right the hybrid catchments (light blue). The bottom panel describes the study sites in terms of topographic areas (left), land cover (center) and anthropic pressure (right).

**For example, authors defined three categories of catchments according to their morphology and geographical locations (lines 148) but it is not clear why only these criteria. It seems that these regions are homogenous also for land use, lithology and climate? Hence, do they belong to the same hydro-ecoregion? It might help to see on figure 1 or in table 1 theses three categories (how many catchments for each category) to link them to geology, land use/land cover.**

Concerning the description of the catchments in Figure 1, please refer to our previous reply. The three categories of catchments are defined for the analysis of the seasonality and the categories are based on the catchment

morphology and geographical locations. Since we analyze the seasonality of in-stream concentrations in relation to the seasonality of discharge, we retain important to differentiate the main hydrological regimes of the catchments, which have a different discharge seasonality. Switzerland is characterized by basically two main geographical zones, the Swiss Plateau, a lowland in the north, and the mountainous Alpine area in the center and south (bottom left panel of Figure 1). The two different zones have substantially different hydrological regimes (Figure S1, upper and bottom panels). However, some of the selected catchments extend in both areas and are therefore defined as "hybrid catchments". These catchments are characterized by a seasonality, which is intermediate between the two extremes (Figure S1, central panel) and they have to be treated separately from the other two classes. The geographical sub-division of these areas is used to differentiate hydrological regimes. It does not imply that these regions are homogeneous in terms of land use or geology. We now present more clearly the catchments characteristics in Figure 1 and we provide further explanations for the analysis of seasonality, pointing out that the classification is done on the basis of the different hydrological regimes and highlighting the main conclusions:

- the seasonality of $Ca^{2+}$, $Na^{2+}$, $K^+$ and $Cl^-$ is dictated by the seasonality of discharge,
- the seasonality of $Mg^{2+}$, TP, DOC and TOC overwhelms the seasonality of discharge due to natural controls,
- the seasonality of $H_4SiO_4$, $NO_3$, TN and DRP overwhelms the seasonality of discharge due to anthropic factors (e.g., input of fertilizers).

We now explain explicitly the criteria of clusterization in L176-185.

**Table 1. Please use km2 as unit for catchment size, and specific discharge (l s-1 km-2) for discharge, also in figures (ex. Figure S5), in order to allow catchment comparisons.**

We thank the reviewer for the suggestion. We adjusted accordingly the unit in Table 1. However, we used specific discharge in mm/h, (mm/h mm/d mm/yr are major units in the hydrological literature) also in Figure 8 and Figure S6.

**ID=VW on table 1 but ID=WM on figure 1. Is it the same catchment? Temporal metrics: it is not very clear what is the aim of each indicator, especially for the seasonality and C-Q relationship.**

Yes, this is the same catchment but the station in the original database changed the name during the monitoring period. The error originates from this inconsistency; we now used only WM throughout the manuscript so that it is consistent. The seasonality indicator, i.e. the index of variability, is used to test the relation between the seasonality of discharge and of concentrations, possibly aiming at isolating the effect of the seasonality of discharge and investigating other factors determining the variability of solute concentrations. We point out the objective of the analysis in L180-183.

C-Q relations, instead, are used as a metric for investigating the behaviors of the different solutes in streams and to investigate whether these behaviors can be generalized across catchments. We added a statement in L205-207: "The empirical relation between solute concentration and discharge $C = a \cdot Q^b$ was explored separately for each solute and for each catchment with the objective of investigating solute behaviors across catchments and whether this behavior can be generalized."

**Index of "seasonal" variability: the numerator of the equation could be reformulated to take into account that it performs a sum of deviations for different catchments belonging to the specific "topographic" class. It is consistent for all "topographic" classes with only 3 to 4 catchments in a category?**

The equation was not properly formulated since it should simply represent the average of the index of seasonal variability over catchments belonging to the same category. We corrected it as indicated here below:

$$Index\ of\ variability = \frac{\sum_n \frac{\left|\frac{\sum_{i=1}^{12} C_i}{\bar{C}} - 1\right|}{\left|\frac{\sum_{i=1}^{12} Q_i}{\bar{Q}} - 1\right|}}{n}$$

**Figure 5. How hydrological regimes were defined? The method is not presented in chapter 3.**

We defined the hydrological regimes based on *Weingartner and Aschwanden*, 1992. We now mentioned this in L176.

**What is the link with Figure 6 (index of seasonal concentration variability), and with figure S2?**

The link between the classes of hydrological regime and Figure 6 (now figure 4) is explained above. The link with Figure S2, instead, refers to one of the results presented in Figure 6 (now Figure 4), i.e., the seasonality of $H_4SiO_4$, $NO_3$, TN and DRP solutes overwhelms the seasonality of discharge likely because of anthropic factors for $NO_3$, TN and DRP (e.g., input of fertilizers). Indeed, these nutrients have their own seasonality, as Figure S2 shows. In the case of $Ca^{2+}$ (bottom panel of Figure S2), instead, the pattern of the load along the year follows quite well the seasonal pattern of discharge, also in the most human-impacted catchments indicating that external forcing is not modifying the seasonality that is due to climate. We emphasized this point in the revised manuscript in L397-400.

**Concentration-discharge relationship. Please define why you calculate integral "b" exponent and truncated "b" exponent, b50sup, b50inf.**

We compute the truncated $b$ exponent (i.e., $b_{50sup}$ and $b_{50inf}$) for the classification of the solute behaviors, because, as explained in lines 216-220, this allows a more insightful classification of their behaviors. The integral $b$, instead was computed to analyze how the anthropic activity influences the solute behavior (Figure 9) beside influencing its magnitude. The objectives of these two parts of the study are different. While the first one aims at the understanding the processes that control the C-Q relations, the second one aims at detecting possible long-term trends in the solute behavior. We specified the use of integral and truncated $b$ in the revised version of the manuscript in L227-230.

**Figure 2 and Figure 10 can be merged, indicating that you use the conceptual diagram of C-Q relationships proposed by Moatar et al, 2017 and test it for Swiss rivers (mean altitudes > 1000 m, mean rainfall 1000 - 2000 mm/y).**

We thank the reviewer for the suggestion and we merged Figure 2 and Figure 10, also simplifying the conceptual diagram defining three main behaviors: biogeochemical stationarity, hydrological dilution and hydrological export:

[Figure]

*Figure 7:* Solute behaviors classification in the log(C)-log(Q) space. The definitions are derived from the classification of *Moatar et al.*, 2017, which is based on the value of $b$, the slope of the regression line in the log(C)-log(Q) space. Discharge time series is divided in low-flow and high-flow events based on $q_{50}$ the median daily discharge. Red areas represent hydrological dilution behavior, yellow areas represent biogeochemical removal for low flows, while green areas represent hydrological export behavior. The grey horizontal line crossing the axes origin represents the near-zero slope area, i.e., it is representative of biogeochemical stationarity. The colorless solutes outside these areas do not show any dominant behavior. The dimension of circles represents the percentage of catchments in which the dominant behavior is observed (from 60 to 100%).

**You can also compare with other recent papers (ex. Diamond, Cohen, 2017 for coastal Plain Rivers in Florida).**

We thank the reviewer for the suggestion and we integrated the suggested reference (and some others) in the manuscript.

**In the split-hydrograph method, separate concentration-discharge relationships are described for below and above median discharge, Q50 is the median of daily discharge. Are your C-Q diagrams (Figure 9, 10) realized from mean bi-monthly concentration with mean bi-monthly discharge? It would be the reason why only 29% of the catchment-solute combinations have different behaviors between low- and highflow conditions.**

The C-Q diagrams are computed from bi-monthly concentration with bi-monthly average discharge. Concentrations are flow-proportional, while the discharge is averaged on a 14-days period. The reviewer raises a very important point. The answer to the question is challenging since it is not possible to have a precise evaluation of which main factor determines different behaviors of solutes between low- and high-flows in 29% of the cases, however we agree that the low sampling frequency may play a role in this. As *Stelzer and Likens* (2006) point out, sampling frequency has different effects depending on the response that concentration has to discharge, so the uncertainty related to the sampling frequency might be different from solute to solute and it is impossible to quantify it with the data available in this study. We added a statement about the influence of sampling frequency in the discussion section. L510-516.

**Or perhaps, it is a characteristic of alpine rivers where dilutions and exports of elements are the major behaviors while biogeochemical and retention removal processes at low flows are not very significant. Or perhaps, Q50 is not the appropriate discharge percentile break-point?**

We did not investigate the effects of using other metrics than $q_{50}$ as break-point, but we discussed this issue in Section 5.3 and cite the suggested reference *Diamond and Cohen,* 2017 where different break-point metrics are tested. However, we think that the characteristics of alpine rivers indeed dampen the role of biological retention and removal processes at low flow, which are therefore not very significant in comparison to other climate and geomorphologies.

**Figure 8. What site? Figure 6. A, B, C not defined in section 3.2**

The site is not mentioned because this is an example of the most common patterns across all of the catchments and we intentionally did not want to refer to any specific catchments. We added in the caption the sites, the patterns are referred to, but we do not think it is an important information for the subsequent discussion since we are discussing general statements. The revised caption thus reads: "***Figure 5***: Three exemplary long-term patterns of solute concentrations. The upper box represents a clear increasing trend, the middle box a non-monotonic trend (firstly increasing and then decreasing), while the bottom box shows the absence of any trend. The patterns refer to the station of Aare – Brugg."

**Figure 10. define grey areas**

We thank the reviewer for the observation and we completed the caption of Figure 10 (now Figure 7) with the definition of the grey areas.

**References**

Diamond, J. S., and Cohen, M J.: Complex patterns of catchment solute-discharge relationships for coastal plain rivers, Hydrological Processes, 32(3), 388-401, 2017.

Godsey, S. E., Kirchner, J. W., and Clow, D. W.: Concentration-discharge relationships reflect chemostatic characteristics of US catchments, Hydrological Processes, 23(13), 1844-1864, 2009.

Huss, M and Fischer, M.: Sensitivity of very small glaciers in the Swiss Alps to future climate change, Front. Earth Sci., 2016.

Ito, M., Mitchell, M. J., Driscoll, C. T., Newton, R. M., Johnson, C. E., and Roy, K. M.: Controls on surface water chemistry in two lake-watersheds in the Adirondack region nitrogen solute of New York: differences in sources and sinks, Hydrological Processes, 21(10), 1249-1264, 2007.

Kaste, O., Stoddard, J. L., and Henriksen, A.: Implication of lake water residence time on the classification of Norwegian surface water sites into progressive stages of nitrogen saturation, Water Air Soil Pollution, 142, 409-424, 2003.

Moatar, F., Abbott, B. W., Minaudo, C., Curie, F., and Pinay, G.: Elemental properties, hydrology, and biology interact to shape concentration-discharge curves for carbon, nutrients, sediment, and major ions, Water Resources Research, 53, 1270–1287, doi:10.1002/2016WR019635, 2017.

Spearman, C.: The proof and measurement of association between two things, The American Journal of Psychology, 1904.

Stelzer, R. S. and Likens, G. E.: Effects of sampling frequency on estimates of dissolved silica export by streams: The role of hydrological variability and concentration-discharge relationships, 2006.

Weingartner, R., and Aschwanden, H.: Abflussregimes als Grundlage zur Abschätzung von Mittelwerten des Abflusses. in: Gruppe für Hydrologie, Universität Bern: Hydrologischer Atlas der Schweiz. Berne: Landeshydrologie, Bundesamt für Wasser und Geologie, plate 5.2, 1992.

Wurtsbaugh, W. A., Baker, M. A., Gross, H. P., and Brown P. D.: Lakes as nutrient "sources" for watersheds: a landscape analysis of the temporal flux of nitrogen through sub-alpine lakes and streams, Verh. Internat. Verein. Limnol., 29, 645-649, 2005.

Zobrist, J.: Water chemistry of Swiss Alpine rivers, in Alpine Waters, edited by: Bundi, U., Springer, Berlin, Heidelberg, 95–118, 2010.

Zobrist, J., Schoenenberger, U., Figure, S., and Hug, S. J.: Long-term trends in Swiss rivers sampled continuously over 39 years reflect changes in geochemical processes and pollution, Environmental Science and Pollution Research, 25:16788-16809, 2018.

---

## Referee Report (RR1)

**General impression**

Botter et al. examines in the resubmitted manuscript "Anthropogenic and catchment characteristic signatures in the water quality of Swiss rivers: a quantitative assessment" the dataset of the Swiss National River and Survey Program (NADUF). The revised version gained clarity on some aspects but also still contains some points to improve.

Different inconsistencies and errors in referring to figures or findings exist and confuse when reading the manuscript (e.g. L391 but also others).

As mentioned in the comments to the first version of the manuscript, a geological map is necessary and would help to link observations and catchment characteristics. L148 mentions "crystalline silicic rocks are dominant" but it seems that the HA or DI catchments have mainly sedimentary rock. Including this in the results and discussion would certainly help.

A methodologically concern is how the parameter $a$ was treated. This was never explained and it is therefore not clear whether this parameter was fixed or variable and could potentially affect conclusions concerning the temporal variable parameter $b$. Please clarify this point in the manuscript.

The comment to the definition of "hybrid catchments" was previously made to stimulate the thinking of the authors about the reason why these catchments are hybrid. My interpretation about the meaning is that they are defined "hybrid" because they contain one or multiple lakes, which could cause a dampened run off, residence of water particles and in or decrease of certain solute concentrations. This is also visible in the different figures. Please discuss the effect of lakes in your study.

The discussion is long and contains a lot of information. One would expect that the analysis of water collected in a human influenced catchment would show an anthropogenic signal. However with the rich dataset and analysis already performed, it is possible to link the different results in a logic way and will allow the reader to understand which are the effects of the climatological forcing (temporal variable signal), catchment characteristics (difference between plateau, alpine and "damped" catchments with lakes) and human influence (specific solutes and their temporal variably). Subsequently it is possible to see different catchment with different runoff regimes and temporal solute behaviour in relation to each and allows highlighting differences and why certain observations or patterns were observed or not. By ending each discussion section with a short summary could help to make stronger statements and highlight the main findings. A summarizing paragraph at the end of the discussion would also help to go beyond a statement that solutes are human induced and write a more specific abstract and conclusion.

These modifications are of minor type. Incorporating the general and specific comments will improve the manuscript and better highlight the specific novel aspects of the manuscript.

**Specific comments referring to line**

Sometimes basins were used while other times 11 catchments. Please use consistent wording for all definitions throughout the manuscript.

L23    "certain solutes" add which ones.

L25    Which variability temporal or spatial is meant here? Which one is higher, the natural or the anthropogenic? The conclusion L587-590 is much clearer.

L106    The Erlenbach and Lumpernenbach seem not really river basins. Please classify them differently!

L107    Which are the subcatchments? Please specify.

L129    Please specify which are the detection threshold and accuracy of the different instruments are. This will help to better understand the signal. I.e. if the instrument accuracy in solute X is ±0.1 and the signal variability as well as, there is not a significant difference between the different catchments (Figure 2). This example is valid for all other figures. Please take this into account and modify accordingly.

L148    Please include a geological map similar to map made by v. Freyberg 2018.

L156    What is a low intensity fertilization, please specify the tons ha$^{-1}$.

L161    Please add after "south-north gradient" and within the alpine valleys.

L164-166    Move to introduction

L177    Maybe you could use a classification: basin (Biggest scale - dampened), catchment ("meso scale"10-... km$^2$) and subcatchment (Small scale <10km$^2$)

L178    What do these extremes relate to, magnitude or timing. Please be specific and use hydrological terms.

| L184-192 | How the index of variability was calculated and presented was confusing. Especially the caption of Figure 4 confuses "*Bar plot of the index of variability. Each bar represents the monthly variability of average concentration 900 relatively to discharge variability per catchment class.*" However, which month was presented in Figure 4, or where different average or spatial average presented here? It's not clear. Please clarify and rewrite this section. |
|---|---|
| L394-399 | S2a and S2b do not correspond with text and makes this section difficult to read. Please modify. |
| L427 | Which analysis and figure support this statement? |
| L434 | Not only fertilizer application but also manure can dissolve calcite and affect Mg/Ca ratios. Please discuss this in the manuscript with appropriate references. |
| L449 | Here the geological map of Switzerland would help. Please also perform a multiple comparison to show significant differences between the different catchments which will help to relate $H_4SiO_4$ to geology. |
| L460 | Is there a reference available to support this statement? Not all Alpine catchments do have high sediment loads (Figure S5). |
| L463 | For clarity refer to the respective figure. |
| L483 | Precipitation amount but also annual distribution are important. When describing precipitation e.g. table 1 it is not clear how this was calculated and also if spatial variability was incorporated e.g. max and min mm $y^{-1}$. |
| L491-493 | By using residence time runoff flow paths are inferred and not demonstrated. Please modify. |
| L506 | "Removal" and entrain are rather similar. Is the word "removal" linked to biological reduction e.g. nitrification? Please specify in the manuscript. |
| L518-520 | This sentence confuses since earlier $H_4SiO_4$ was linked to weathering and geology (L303) while now to bioactive processes. Please clearly state which is the dominant process. |
| L526 | Please specify here which catchments are mentioned here. |
| L528 | But fast response is followed by fast recession, are alpine and sub-alpine rather affected by dilution caused by the high precipitation? |
| L532 | This statement is not fully clear. Isn't it that the water "picks" up a certain composition due to its flow-path? |
| L567 | Statement concerning subsurface flow is not clear. Is subsurface flow only occurring with natural conditions while human activity influences overland flow? Please modify this sentence |
| L588 | All or only certain solutes?

What is the anthropogenic signal, one solute or a series? What is the influence of the catchment characteristics? Do small catchments with high agriculture % behave different solutes compared to large catchments with lakes? Although statistically it is difficult to show relations between observations and catchment characteristics, a quantitative statement can be made to explain certain observations. Also, explain what caused the temporal variability in Figure 5, 6 and 8. Clearly highlight in your take home message e.g. there is a variability due to climate but anthropogenic can be noticed in region 1 while less in catchments in region 2 because of ... |
| L594 | What is a "macro-pattern"? Please explain or remove. |
| L604 | Important findings are missing e.g. a non-seasonal behaviour of "anthropic" induced solutes as e.g. discussed in L395 and differences between different catchments are missing. |
| L606-608 | This statement is too general and not fully clear. |
| L614 | A clear take home message of how the different regions or catchment types are missing and would be valuable to include. |
| Figure 1 | catchment boundaries are difficult to see. From the stream network it seems that the hybrid catchments containing lakes, are dampened. Maybe just call them dampened catchment. This would also be agreement of Figure S2. Also add letters a-... to facilitate connecting the text and different panels. |
| Figure 5 &S2 | The variability in time is interesting. Is it possible to relate this temporal pattern to climate variability, i.e. wetter and drier decades? In addition is it possible to state that in wetter years the anthropogenic signal is higher or lower (e.g. Mg, $No_3$ or …) ? Please include in manuscript. |
| Figure 8b | Really interesting to see changes but which variable do I see…? Continue to label the different panels as letters or add as new figure. |

---

## Author Response (AR2)

**Reply to Reviewer 1**

**General impression**

**Botter et al. examines in the resubmitted manuscript "Anthropogenic and catchment characteristic signatures in the water quality of Swiss rivers: a quantitative assessment" the dataset of the Swiss National River and Survey Program (NADUF). The revised version gained clarity on some aspects but also still contains some points to improve.**
We thank the reviewer for the positive comment to the revised version of the manuscript and for highlighting points where the manuscript can be improved.

**Different inconsistencies and errors in referring to figures or findings exist and confuse when reading the manuscript (e.g. L391 but also others).**
We thank the reviewer for pointing out these inconsistencies and we refers him/her to the specific comments for the corrections.

**As mentioned in the comments to the first version of the manuscript, a geological map is necessary and would help to link observations and catchment characteristics. L148 mentions "crystalline silicic rocks are dominant" but it seems that the HA or DI catchments have mainly sedimentary rock. Including this in the results and discussion would certainly help.**
We added to Figure 1 the geological map and we referred to it throughout the manuscript. Please see also the specific comments.

**A methodologically concern is how the parameter *a* was treated. This was never explained and it is therefore not clear whether this parameter was fixed or variable and could potentially affect conclusions concerning the temporal variable parameter *b*. Please clarify this point in the manuscript.**
The *a* parameter is a coefficient with units of concentration resulting from the linear interpolation of discharge (Q) and concentration (C) data. It is computed together with *b* but it does not affect *b*. Being our analysis focused on the solute behaviours, which, according to the wide literature, is determined by the *b* exponent, the *a* coefficient is unimportant. However, we agree that we should mention in the manuscript how it is computed. We added a statement in Paragraph 3.2 (L214-216).

**The comment to the definition of "hybrid catchments" was previously made to stimulate the thinking of the authors about the reason why these catchments are hybrid. My interpretation about the meaning is that they are defined "hybrid" because they contain one or multiple lakes, which could cause a dampened run off, residence of water particles and in or decrease of certain solute concentrations. This is also visible in the different figures. Please discuss the effect of lakes in your study.**
We agree and we explain that the catchments we define "hybrid" are characterised by extended surface. They all include lakes in their domain, but also other catchments that we classified as Alpine or in the Swiss Plateau area have comparable percentage of lakes in their domain, e.g., BR and WM have the same fraction of lake surface in their domain. The subdivision per macro-geological areas is more functional to the analysis we carried out. However, we agree with the Reviewer that discussing the issue of lakes in the manuscript is necessary. We therefore added some statements at L180-185.

**The discussion is long and contains a lot of information. One would expect that the analysis of water collected in a human influenced catchment would show an anthropogenic signal. However with the rich dataset and analysis already performed, it is possible to link the different results in a logic way and will allow the reader to understand which are the effects of the climatological forcing (temporal variable signal), catchment characteristics (difference between plateau, alpine and "damped" catchments with lakes) and human influence (specific solutes and their temporal variably). Subsequently it is possible to see different catchment with different runoff regimes and temporal solute behaviour in relation to each and allows highlighting differences and why certain observations or patterns were observed or not. By ending each discussion section with a short summary could help to make stronger statements and highlight the main findings. A summarizing paragraph at the end of the discussion would also help to go beyond a statement that solutes are human induced and write a more specific abstract and conclusion.**
We thank the Reviewer for the useful suggestion and we modified the discussion. Specifically, we focused on the first part of the discussion, which contains a large amount of information and we introduced a final outlook to sections 5.1 and 5.2 to summarise the take-home messages. Section 5.3 was better structured and the take-home messages are already specified in the conclusion, so that an outlook would have been redundant. Please

note that key messages are summarized in a short conclusion section, repeating those a few lines earlier in the discussion would be excessively redundant in our opinion.

**These modifications are of minor type. Incorporating the general and specific comments will improve the manuscript and better highlight the specific novel aspects of the manuscript.**
We thank the Reviewer for the general positive impression about our work and for the suggestions that contributed to improving the manuscript.

**Specific comments referring to line**

**Sometimes basins were used while other times 11 catchments. Please use consistent wording for all definitions throughout the manuscript.**
In the revised version of the manuscript we use consistently the term "catchments".

**L23 "certain solutes" add which ones.**
We explicitly point out the solutes in the revised version of the manuscript.

**L25 Which variability temporal or spatial is meant here? Which one is higher, the natural or the anthropogenic? The conclusion L587-590 is much clearer.**
In this case we refer to the variability in time, which we tackle in the seasonality analysis. We modified with the explicit expression "The variability in time..". We also underlined the stronger variability due to anthropogenic factors adding the statement "..and the most significant trends in time are due to the variation of the anthropogenic forcing in the long-term".

**L106 The Erlenbach and Lumpernenbach seem not really river basins. Please classify them differently!**
We adjusted the definition of case studies at L107-108 as follows: "The resulting case studies include 5 catchments (Thur - AN, Aare - BR, Rhine – WM, Rhone – PO and Inn - SA), 3 sub-catchments (Rhone – PO, Rhine – RE and Rhine – DI) and 2 small headwater catchments (Erlenbach and Lümpenenbach)."

**L107 Which are the subcatchments? Please specify.**
 Please, see comment above.

**L129 Please specify which are the detection threshold and accuracy of the different instruments are. This will help to better understand the signal. I.e. if the instrument accuracy in solute X is ±0.1 and the signal variability as well as, there is not a significant difference between the different catchments (Figure 2). This example is valid for all other figures. Please take this into account and modify accordingly.**
We added a short paragraph in the Supplementary Information (Paragraph S1) focused on the detection thresholds issue and we commented on this in relation to the results.

**L148 Please include a geological map similar to map made by v. Freyberg 2018.**
We thank the reviewer for the suggestion and we included it in Figure 1.

**L156 What is a low intensity fertilization, please specify the tons ha-1.**
We thank the reviewer for this comment because we realised the previous version of the definition of "low intensity fertilization" was incorrect. We rectified the definition of extensive agriculture in the revised version of the manuscript at L157-158.

**L161 Please add after "south-north gradient" and within the alpine valleys.**
We added this information to the revised version of the manuscript (L163-164).

**L164-166 Move to introduction**
We agree with the Reviewer that this statement is not in the correct section and we removed it from the manuscript, since in the introduction it was redundant.

**L177 Maybe you could use a classification: basin (Biggest scale - dampened), catchment ("meso scale"10- ... km2) and subcatchment (Small scale <10km2)**
We adjusted the catchments definition in L107-108. Here, instead, we keep the morphology-based classification because, we think it is more functional to the following analyses.

**L178 What do these extremes relate to, magnitude or timing. Please be specific and use hydrological terms.**

We thank the reviewer for this observation and we added a more precise hydrological description at L178-179.

**L184-192 How the index of variability was calculated and presented was confusing. Especially the caption of Figure 4 confuses "*Bar plot of the index of variability. Each bar represents the monthly variability of average concentration 900 relatively to discharge variability per catchment class.*" However, which month was presented in Figure 4, or where different average or spatial average presented here? It's not clear. Please clarify and rewrite this section.**

We clarified in the caption of Figure 4 that the bar plot refers to the average monthly deviation and not to the deviation of a specific month. To enhance clarity, we added to the manuscript a non-symbolic formulation of the expression of the index of variability (L194).

**L394-399 S2a and S2b do not correspond with text and makes this section difficult to read. Please modify.**

We apologise for the inconsistency and we corrected the reference to the figure from Figures S2a and S2b to Figures S3a and S3b.

**L427 Which analysis and figure support this statement?**

The analysis of trends is supporting this result. In the revised version of the manuscript we are more precise referring explicitly to the analysis and to the Figure showing the result.

**L434 Not only fertilizer application but also manure can dissolve calcite and affect Mg/Ca ratios. Please discuss this in the manuscript with appropriate references.**

Manure can be somehow considered a fertilizer. Nonetheless, we thank the reviewer for the suggestion. We integrated this aspect in the discussion with appropriate references (L443-444)

**L449 Here the geological map of Switzerland would help. Please also perform a multiple comparison to show significant differences between the different catchments which will help to relate H4SiO4 to geology.**

We now referred to the new geological map introduced in Figure 1.

**L460 Is there a reference available to support this statement? Not all Alpine catchments do have high sediment loads (Figure S5).**

Accordingly to the catchment classification in L111-112, in the revised version of the manuscript we explicitly say that the results are supported by the analysis for all the Alpine catchments except for the two small headwater catchments, because, with their small size, they are peculiar.

**L463 For clarity refer to the respective figure.**

We added the reference to Figure 4.

**L483 Precipitation amount but also annual distribution are important. When describing precipitation e.g. table 1 it is not clear how this was calculated and also if spatial variability was incorporated e.g. max and min mm y-1.**

All the description data come from the information included in the NADUF database, which provides a satisfactory description of each catchment. The data source was not clear from the caption of Table 1 so we modified the caption to make it clearer.

**L491-493 By using residence time runoff flow paths are inferred and not demonstrated. Please modify.**

We apologise for the incorrect citation and modified accordingly.

**L506 "Removal" and entrain are rather similar. Is the word "removal" linked to biological reduction e.g. nitrification? Please specify in the manuscript.**

Yes, we meant exactly this kind of removal processes. We modified to enhance clarity.

**L518-520 This sentence confuses since earlier H4SiO4 was linked to weathering and geology (L303) while now to bioactive processes. Please clearly state which is the dominant process.**

We thank the reviewer for pointing out the inconsistency. We clarified this point.

**L526 Please specify here which catchments are mentioned here.**

We specified the catchments.

**L528 But fast response is followed by fast recession, are alpine and sub-alpine rather affected by dilution caused by the high precipitation?**
A high precipitation is definitely an important reason for the fast hydrological response (e.g., fast peak flows, fast recession). We mentioned explicitly this point in the text. However, high-precipitation alone would not be a sufficient reason, and dilution should be also related to the fact that water already in the catchment and the new water may not mix properly. As a matter of fact, in a groundwater dominated system, with large groundwater volumes, the chemical composition of discharge is expected to be more similar and to show little dilution regardless of the amount of precipitation.

**L532 This statement is not fully clear. Isn't it that the water "picks" up a certain composition due to its flow-path?**
It depends from which perspective one looks at the problem, if from the perspective of a parcel of water entering the catchment and following a certain flow-path (in this case, yes, it "picks" a certain composition). If we look at the problem from the perspective of some water leaving the catchment from an outlet. In this case, the chemical composition is a combination of the chemistry originated by the multiple flowpaths reaching the outlet at that time. This second case is what is measured by water-quality observations and it is what we refer to in the discussion. We acknowledged the fact that complex watershed in the Alpine region may have multiple flowpaths, including very long ones, but we state that given the widespread observed dilution behaviour those flowpaths are unlikely dominant in the composition of the outlet discharge.

**L567 Statement concerning subsurface flow is not clear. Is subsurface flow only occurring with natural conditions while human activity influences overland flow? Please modify this sentence**
We rephrased the sentence to improve clarity. Subsurface flow is occurring regardless of the fact the catchment is natural or affected by human activities, it is the effect on DOC, which can be seen only if the catchment is natural, because in catchments affected by human activities the DOC signature is not anymore due only to natural processes.

**L588 All or only certain solutes?**
    **What is the anthropogenic signal, one solute or a series? What is the influence of the catchment characteristics? Do small catchments with high agriculture % behave different solutes compared to large catchments with lakes? Although statistically it is difficult to show relations between observations and catchment characteristics, a quantitative statement can be made to explain certain observations. Also, explain what caused the temporal variability in Figure 5, 6 and 8. Clearly highlight in your take home message e.g. there is a variability due to climate but anthropogenic can be noticed in region 1 while less in catchments in region 2 because of ...**
We agree with the Reviewer that the conclusion was general and sometimes vague. In the revised version of the manuscript we integrated some more detailed messages referring to the explanation of results pointed out by the Reviewer.

**L594 What is a "macro-pattern"? Please explain or remove.**
We removed the sentence.

**L604 Important findings are missing e.g. a non-seasonal behaviour of "anthropic" induced solutes as e.g. discussed in L395 and differences between different catchments are missing.**
We thank the reviewer for the suggestion and added some statements to the conclusion.

**L606-608 This statement is too general and not fully clear.**

We rephrased this part.

**L614 A clear take home message of how the different regions or catchment types are missing and would be valuable to include.**
We are not sure to fully understand what the reviewer meant with this comment, however, we prefer to conclude the manuscript with what we can clearly state rather than what is missing from the analysis.

**Figure 1 catchment boundaries are difficult to see. From the stream network it seems that the hybrid catchments containing lakes, are dampened. Maybe just call them dampened catchment. This would also be agreement of Figure S2. Also add letters a-... to facilitate connecting the text and different panels.**

We modified Figure 1 as suggested. Particularly:
- The map with the morphological regions was substituted with the maps of macro-geological classes because it is more informative;
- Borders of catchments were highlighted;
- The letters a, b (upper panel) and c, d, e (bottom panel) were added

For comments to the definition "dampened catchments" please refer to the answer above.

**Figure 5 &S2 The variability in time is interesting. Is it possible to relate this temporal pattern to climate variability, i.e. wetter and drier decades? In addition is it possible to state that in wetter years the anthropogenic signal is**

**higher or lower (e.g. Mg, No3 or …) ? Please include in manuscript.**

In the first round of Review, Referee 1 raised this point, therefore we considered the natural variability of climate as a potential driver for the long-term trends that we observe in the concentration data. The results are shown in the reply to the Referee, but the conclusion are already stated in L200-207. Consistently with this statement, we did not investigate the relationship between the climatic and the anthropogenic forcing.

**Figure 8b Really interesting to see changes but which variable do I see…? Continue to label the different panels as letters or add as new figure.**

We adjusted Figure 8 adding the name of the solute in the capture and adding the label to the left panel.